# Genetic polymorphisms lead to major, locus-specific, variation in piRNA production in mouse

Eduard Casas [1,2,3,12], Adrià Mitjavila-Ventura [1,2,4,12], Pío Sierra [5,12], Cristina Moreta-Moraleda [6,7], Judith Cebria[8], Ilaria Panzeri [9], J Andrew Pospisilik [9], Josep C Jimenez-Chillaron[7,8,10], Sonia V Forcales [6,11✉] & Tanya Vavouri [1,2,5✉]

## Abstract

PIWI-interacting RNAs (piRNAs) are small noncoding RNAs that silence transposons in the animal germline. PiRNAs are produced from long single-stranded noncoding transcripts, from protein-coding transcripts, as well as from transposons. While some sites that produce piRNAs are in deeply conserved syntenic regions, in general, piRNAs and piRNA-producing loci turnover faster than other functional parts of the genome. To learn about the sequence changes that contribute to the fast evolution of piRNAs, we set out to analyze piRNA expression between genetically different mice. Here we report the sequencing and analysis of small RNAs from the mouse male germline of four classical inbred strains, one inbred wild-derived strain and one outbred strain. We find that genetic differences between individuals underlie variation in piRNA expression. We report significant differences in piRNA production at loci with endogenous retrovirus insertions. Strain-specific piRNA-producing loci include protein-coding genes. Our findings provide evidence that transposable elements contribute to inter-individual differences in expression, and potentially to the fast evolution of piRNA-producing loci in mammals.

**Keywords** ERV; Diversity; Mouse; piRNA; PIWI
**Subject Categories** Development; Evolution & Ecology; RNA Biology

## Introduction

Eukaryotic genomes are hosts to a great number and diversity of transposons, which vary between species and between individuals of the same species (for a review see (Cosby et al, 2019). Active transposons are mobile genetic elements that can propagate within the genome of a cell. When a transposon replicates in the germline, the new copy is passed on to the next generation. To counter potentially deleterious, heritable, mutagenic events caused by transposons, living organisms have evolved mechanisms that repress these elements in the germline. One of the most important defense mechanisms protecting the animal germline against transposons is the PIWI-interacting RNA (piRNA) pathway (reviewed in (Siomi et al, 2011; Ozata et al, 2019)). The core components of the piRNA pathway are deeply conserved and active in the germline of almost all animals. Yet, even closely related mammalian species produce distinct sets of piRNAs. The genetic mechanisms that lead to the fast diversification and divergence of piRNAs between species are largely unknown.

The mammalian piRNA pathway surveys the germline for transposon transcripts and transposon-containing genomic loci using millions of piRNAs, each with a distinct sequence. PiRNAs are small noncoding RNAs produced from a few hundred loci (Gainetdinov et al, 2018). Approximately half of these loci are noncoding, while the other half are protein-coding genes (Li et al, 2013). Why some germline-expressed coding or noncoding transcripts become processed into piRNAs and others do not remains unclear. A mechanism that defines a transcript as a piRNA precursor is the presence of a sequence with extensive complementarity to initiator piRNAs produced from other genomic loci (Gainetdinov et al, 2018). Yet, this requires constant expression of piRNAs throughout the life cycle of the mammalian germline, which is not the case, suggesting that there are additional mechanisms that trigger piRNA production during development and evolution.

The evolution of piRNAs is fast. Some piRNA-producing loci are found in syntenic regions of distantly related species, however their sequence is not conserved (Girard et al, 2006; Chirn et al, 2015). Furthermore, the sites from which piRNAs are produced are more often than not, species-specific (Assis and Kondrashov, 2009; Özata et al, 2020; Chirn et al, 2015). Within the human population, piRNA-producing loci contain more genetic variants than any

[1]Josep Carreras Leukaemia Research Institute (IJC), Ctra de Can Ruti, Camí de les Escoles s/n, Badalona, Barcelona 08916, Spain. [2]Germans Trias i Pujol Research Institute (IGTP), Can Ruti Campus, Badalona, Barcelona 08916, Spain. [3]University of Barcelona (UB), Barcelona, Spain. [4]Autonomous University of Barcelona (UAB), Barcelona, Spain. [5]Open University of Catalunya (UOC), Barcelona, Spain. [6]Department of Pathology and Experimental Therapeutics, School of Medicine and Health Sciences, Campus of Bellvitge, University of Barcelona, Carrer de la Feixa Llarga, s/n, L'Hospitalet de Llobregat, Barcelona, Spain. [7]Oncobell Program, Bellvitge Biomedical Research Institute (Idibell), Gran Via de les Corts Catalanes L'Hospitalet de Llobregat, Barcelona, Spain. [8]Institut de Recerca Sant Joan de Déu, Endocrine Division, Esplugues de Llobregat, Barcelona 08950, Spain. [9]Center for Epigenetics, Van Andel Research Institute, Grand Rapids, MI 49503, USA. [10]Department of Physiological Sciences, School of Medicine, University of Barcelona, Carrer de la Feixa Llarga, s/n, L'Hospitalet de Llobregat, Barcelona, Spain. [11]Serra Húnter Professor, Department of Pathology and Experimental Therapeutics, School of Medicine and Health Sciences, Campus of Bellvitge, University of Barcelona, Carrer de la Feixa Llarga, s/n, L'Hospitalet de Llobregat, Barcelona, Spain. [12]These authors contributed equally: Eduard Casas, Adrià Mitjavila-Ventura, Pío Sierra. ✉E-mail: sforcales@ub.edu; tvavouri@carrerasresearch.org

other transcribed genomic feature (Özata et al, 2020). These independent pieces of evidence suggest that there is little purifying selection pressure on the DNA sequence of piRNA-producing loci. Given that approximately half of all mammalian piRNA precursors are transcripts of protein-coding genes, it is remarkable how evolvable piRNA-producing loci are.

Considering the fast evolution of piRNAs and piRNA-producing genes, how different is the expression of piRNAs in genetically different, healthy individuals of the same species? Differences in expression of piRNAs have been found between human individuals, however, it is unclear whether this is due to differences in health, exposures to genotoxic agents or genetics (Özata et al, 2020). Under controlled environmental conditions, analyses of piRNAs from different Drosophila (Kelleher and Barbash, 2013; Shpiz et al, 2014) and zebrafish strains (Kaaij et al, 2013) have revealed that the identity of piRNA-producing loci and their expression levels vary depending on their genetic background. In Drosophila, the comparison of two strains revealed that transposable element insertions in euchromatic regions induce the formation of dual-strand piRNA-producing loci at the site of insertion and single-strand piRNAs downstream of transposable element insertions in 3' UTRs of protein-coding genes (Shpiz et al, 2014). Although the piRNA pathway is conserved between flies and mice, some aspects of piRNA biogenesis are distinct, such as how piRNA-producing loci are marked in the genome and how they are transcribed (e.g., (ElMaghraby et al, 2019; Kneuss et al, 2019; Yu et al, 2021)). Even though piRNAs have been extensively studied in mice and play an essential role in male fertility, it remains to be determined whether there are differences in the loci that produce piRNAs in different mice.

We sought to quantify the variation in piRNA expression in different strains of mice and then use it to search for potential genetic mechanisms for this variation. We sequenced and analyzed small RNAs from the male germline of 57 adult mice from four classical inbred mouse strains (C57BL/6J, 129S1/SvImJ, C3HeB/FeJ, NOD), one wild-derived inbred strain (CAST/EiJ) and one outbred strain (ICR). We found significant differences in piRNA production from different genomic loci between genetically diverse mice and only minimal differences between mice of the same inbred strain. We tested the link between variation in piRNA expression and transposable element insertions or deletions and found a highly significant association, specifically for the murine endogenous retrovirus (ERV) Intracisternal A particle (IAP). Taken together with the previous work in fruitflies, our work in mice reveals that new transposable element insertions are a deeply conserved genetic mechanism for piRNA diversification within a species and the emergence of new piRNA-producing loci during evolution.

# Results

## Variation in piRNA expression between genetically diverse mice

We set out to analyze inter-strain variation in small RNA production from known piRNA-producing loci (also known as piRNA clusters) of the mouse genome, aiming to understand the level of piRNA expression variation between genetically diverse individuals in a mammalian species. As a first approach, we studied inter-individual variation in abundance of small RNAs mapped to 214 known piRNA-producing loci (Li et al, 2013) from RNA extracted from whole testes of young adult mice from four classical inbred strains: C57BL/6J (referred to as BL6), NOD, C3HeB/FeJ (referred to as C3H) and 129S1/SvImJ (referred to as 129) (Fig. 1; Datasets EV1–3). The majority (72–80%) of small RNAs sequenced from whole testis from all four inbred strains mapped to known piRNA clusters (Dataset EV1). For brevity, we refer to small RNAs mapping to known piRNA clusters as piRNAs, even though these small RNAs were not identified as bound to PIWI proteins by us. The reference set, which corresponds to BL6 (Li et al, 2013), contains 84 loci producing piRNAs predominantly during the prepachytene stages of meiosis, 100 loci producing piRNAs predominantly from the pachytene stage of meiosis onwards and 30 loci that are known as hybrid, expressed throughout adult spermatogenesis (Li et al, 2013). As expected, the loci with the highest count of piRNAs in the whole adult testis in all four mouse strains are the pachytene ones (Appendix Figs. S1 and S2). Overall, piRNA abundance was highly correlated between animals of the four classical inbred strains (Fig. 1A,B).

Despite the high concordance of piRNA expression between all the studied mice, the abundance of piRNAs was more similar between animals of the same inbred strain than animals of different strains (Fig. 1B; Appendix Fig. S3). To assess statistically the effect of the strain on global piRNA cluster expression, we used a linear model with strain as a fixed effect and piRNA cluster as a random effect and found that strain significantly contributes to piRNA cluster expression level ($P$ value = 0.04). Considering each cluster independently, strain significantly contributes to inter-individual variation in piRNA cluster expression in 159 out of 192 (83%) clusters (Dataset EV4). Notably, strain explains at least 80% of inter-individual variation in expression in 127 out of 192 (66%) the clusters (Fig. 1C). We conclude that in classical inbred strains, for a subset of piRNA clusters, genetics is the most important factor affecting inter-individual variation of piRNA expression.

In pairwise comparisons of the inbred mouse strains, of the 214 known piRNA clusters, 21 were significantly differentially expressed in at least one pairwise comparison and eleven of these clusters were significantly differentially expressed in at least three of the six pairwise comparisons between the four inbred mouse strains (Fig. 1A; Dataset EV5). Differentially expressed piRNA-producing loci include protein-coding genes, such as *Noct* (also previously known as *Ccrn4l*), *Zbtb37* and *Mrs2* and noncoding cluster 10-qC1-2617 on chromosome 10, among others (Fig. 1E). Also, differentially expressed clusters include deeply conserved as well as mouse-specific clusters. The protein-coding gene *Nocturnin* (*Noct*) is a mouse-specific prepachytene piRNA cluster (Li et al, 2013) that produces abundant piRNAs in BL6 and NOD strains but not in C3H and 129 (Fig. 1E,F; Appendix Figs. S2 and S4A). The gene *Zbtb37*, a prepachytene piRNA cluster that also produces piRNAs in human, macaque and platypus, produces significantly more piRNAs in strains NOD, C3H and 129 than in BL6 (Fig. 1E; Appendix Figs. S2 and S4B). The gene *Mrs2*, which is a hybrid, mouse-specific piRNA cluster, produces piRNAs in three mouse strains but nearly none in NOD (Fig. 1E; Appendix Figs. S2 and S4C). Last, an intergenic, pachytene, cluster on chromosome 10 that is not conserved in other species, produces few piRNAs in strain 129, many piRNAs in strains NOD and C3H and still twice as many in BL6 (Fig. 1E; Appendix Figs. S2 and S4D). Thus, select

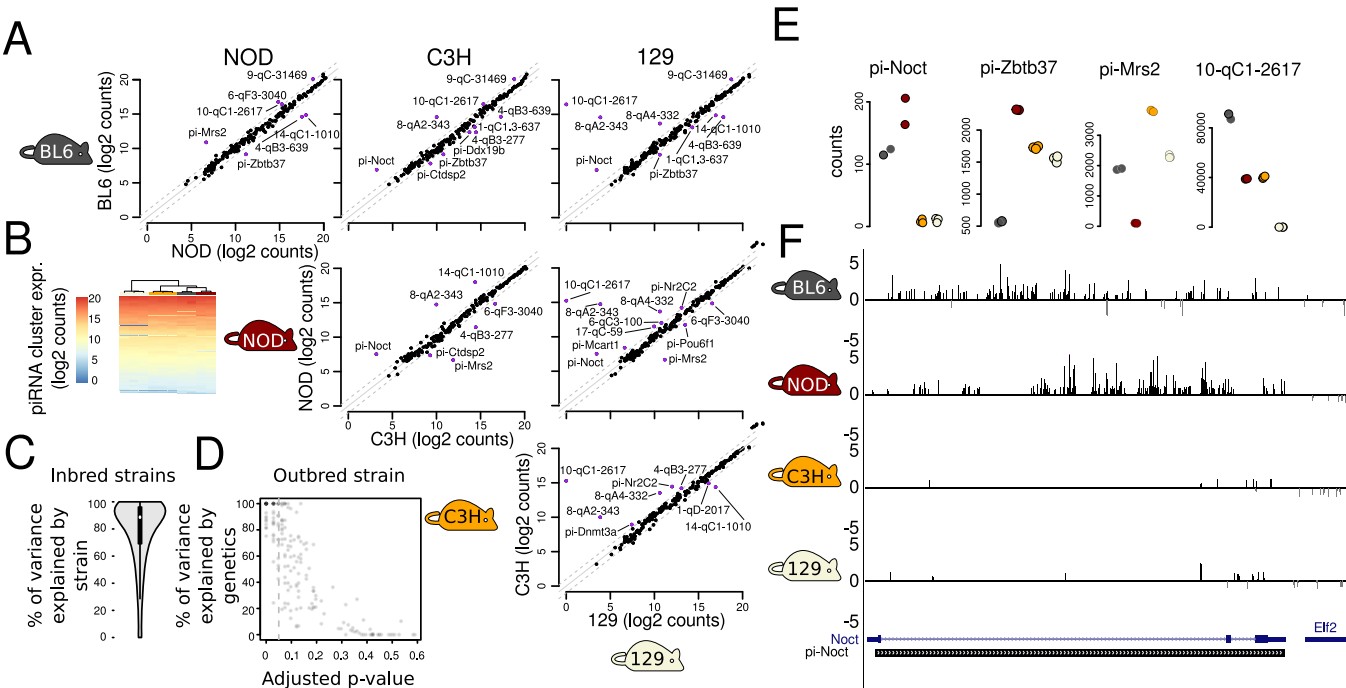

**Figure 1. Variation in expression of 214 previously defined piRNA-producing loci in testes of four classical inbred mouse strains.**

(A) Pairwise comparison of expression of 214 previously defined piRNA-producing loci in four classical inbred mouse strains. Significantly differentially expressed piRNA clusters are shown using purple points, and their names are shown. Each point represents the mean of the log2 transformed, scaled read counts of all samples of the same strain (3 samples each, for strains C3H and 129 and 2 samples each, for strains NOD and BL6). (B) Heatmap of piRNA cluster expression in each mouse sample of the four inbred strains. (C) Distribution of percentage of piRNA cluster expression level variance explained by (inbred) strain. Expression values were VST transformed and the variance explained by genetics calculated using a linear model with strain representing the effect of genetics. The boxplot inside the violin plot shows the interquartile range, the middle point corresponds to the median, and the whiskers extend to the extreme values provided they are no more than 1.5 times the interquartile range from the box. $N = 191$ piRNA-producing loci. (D) Percentage of piRNA cluster expression level variance explained by genetics estimated from mice of the outbred strain ICR in multi-generation pedigrees. Empirical $P$ values are based on 100 permutations and adjusted for multiple testing using the Benjamini–Hochberg method (see "Methods"). (E) Five piRNA-producing loci with highly variable expression in four classical inbred mouse strains. Data points show the scaled small RNA counts with BL6 samples shown in gray, NOD in red, C3H in yellow and 129 in beige. Small RNAs mapped at these five loci on each strand and in each replicate are shown in Appendix Fig. S4. (F) Classical inbred mouse strains produce significantly different levels of small RNAs from pi-Noct (also known as pi-Ccrn4l). Genes and repeats are also shown. One sample from each strain was randomly chosen. The 214 piRNA-producing loci were defined in BL6 (Li et al, 2013).

piRNA clusters produce different steady state levels of piRNAs in mice of different strains.

The differences in piRNA cluster expression between strains prompted us to explore the relationship between genetic differences between strains and variation in piRNA abundance. Using known SNPs between the four inbred mouse strains, for each cluster, we compared the genetic distance tree to the expression distance tree. The genetic and expression trees match for pi-Noct and pi-Mrs2, they match partially for pi-Zbtb37 while the noncoding piRNA cluster 10-qC1-2617 does not contain enough SNPs to calculate a genetic distance tree (see Appendix Fig. S5 and Dataset EV6). Comparing piRNA-producing loci to other genomic regions, between mouse strains, piRNA-producing loci (excluding those also coding for protein-coding genes) have more sequence differences compared to other long noncoding RNAs (Appendix Fig. S6). Focusing on pachytene piRNA-producing loci that are the ones most abundant in whole testes, and comparing BL6 to the other three strains, we found that differentially expressed loci tend to have more sequence differences along the entire locus (Appendix Fig. S7A), at the promoter (Appendix Fig. S7B) and at transcription

factor binding sites in promoters (Appendix Fig. S7C), although there is a low number of observations and not all comparisons reach statistical significance.

To gain independent evidence for the link between genetic variation and gene expression variation, we sequenced and analyzed small RNAs from the testes of 39 young adult mice of the genetically outbred strain ICR (for further details on this dataset, see "Methods" and Datasets EV1 and EV7). These mice were from seven pedigrees (Appendix Fig. S8). If genetics has a major effect on inter-individual piRNA cluster expression, then, at least for some piRNA clusters, closely related individuals should have more similar piRNA cluster expression than less related individuals. For each piRNA cluster, we tested whether relatedness is a factor that significantly explains piRNA expression variation. We found that on average 48% of variation of piRNA cluster expression is explained by relatedness with a distribution skewed towards high values (Fig. 1D; Dataset EV8). Because the dataset is relatively small and the information incomplete (we only have data for males), we used permutations of individuals in pedigrees to assess the significance of the estimates of heritability of piRNA

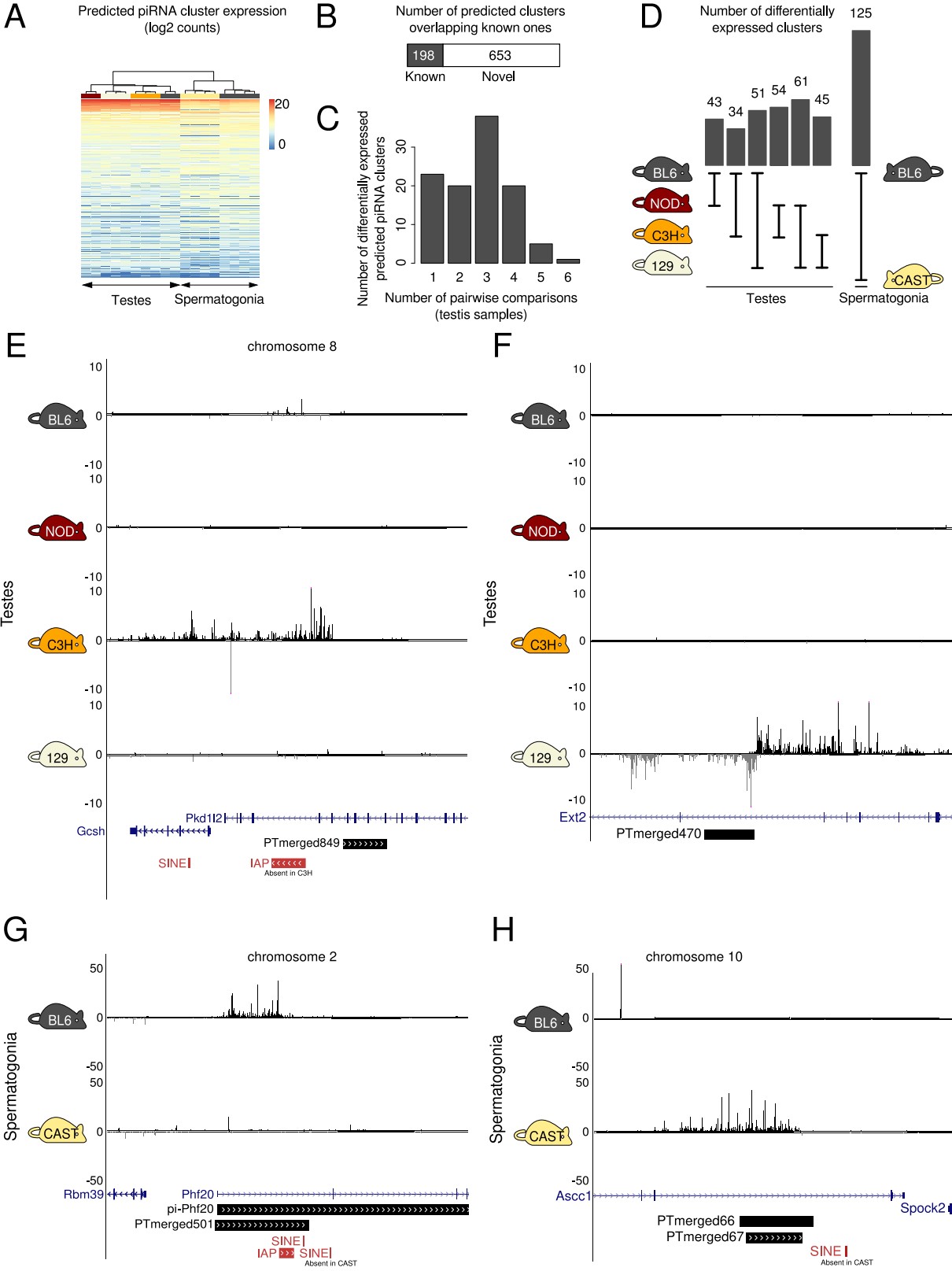

**Figure 2.  Many predicted piRNA-producing loci show significant differences in expression between mouse strains.**

(A) Heatmap showing clustering of expression of predicted piRNA-producing loci from whole testes and isolated spermatogonia from inbred mouse strains. (B) Proportion of predicted piRNA-producing loci overlapping previously known ones from (Li et al, 2013). (C) Frequency distribution of the number of differentially expressed predicted piRNA clusters in testis samples from four mouse strains. (D) Number of differentially expressed predicted piRNA-producing loci per pairwise strain comparison. As there are small RNA data from testes from four strains, there is a total of six pairwise comparisons. (E) Mono-directional predicted piRNA cluster PTmerged849 is highly expressed only in the testes of C3H mice. (F) Bi-directional predicted piRNA cluster PTmerged470 is highly expressed only in the testes of 129 mice. (G) Predicted piRNA cluster PTmerged501 (cluster pi-Phf20 from (Li et al, 2013)) is expressed in BL6 and not in CAST spermatogonia. (H) Predicted piRNA cluster PTmerged66/67 is expressed in CAST and not in BL6 spermatogonia. The positions of genes, repeats and multi-strain alignments from the UCSC Genome Browser are also shown.

cluster expression. For each piRNA cluster, we compared the variance explained by genetics calculated from individuals in their true pedigrees against that from 100 permutations of the pedigrees. We found that genetics significantly contributes to at least 50% of the variation in piRNA cluster expression for 87 out of the 214 clusters (40.7%). We wondered whether genetic differences and different degrees of relatedness also explain variation in piRNA cluster expression for the top five clusters that vary the most in this set. For three out of five of these piRNA clusters, genetics significantly explains most of the inter-individual piRNA cluster expression (Appendix Fig. S9). The results from both inbred and outbred mice, collectively demonstrate that for some piRNA clusters, genetics explains most of the inter-individual expression variation.

We reasoned that there are likely additional piRNA clusters with significant expression differences between strains that are so far missed because they are not expressed in the BL6 strain. To address this bias for the reference mouse strain, we used the testis small RNA data from all four strains (BL6, NOD, C3H, and 129) as well as eight samples from isolated spermatogonia from the reference mouse strain (BL6) and the wild-derived inbred mouse strain CAST/EiJ (referred to as CAST) to predict clusters de novo (see "Methods"). After predicting clusters in each strain genome, we merged them into one set, discarding those that are not in alignable regions (usually gaps in strain assemblies). There are 900 predicted piRNA-producing loci, 82% of which produce piRNAs from only one strand and 18% from both strands. Of the predicted clusters, 198 overlap known mouse-piRNA clusters while the rest are novel (Fig. 2B). We compared the expression of these predicted clusters between mouse strains (Fig. 2). As expected, predicted piRNA clusters have more similar expression between samples of the same strain (Fig. 2A; Appendix Fig. S3B). Also, piRNA cluster expression of samples from testis (enriched for pachytene piRNAs) and spermatogonia (enriched for prepachytene piRNAs) cluster separately (Fig. 2A). Among the 900 predicted piRNA clusters, we found 107 that are differentially expressed in testis samples of the four strains. Thirty-eight of these clusters are differentially expressed in testes of one of the four strains (i.e., significant in three pairwise comparisons) (Fig. 2C). The samples from BL6 and CAST spermatogonia had the highest number of differentially expressed predicted clusters, likely because CAST is genetically more different from the rest of the classical inbred strains (Fig. 2D). In total, we found 332 clusters differentially expressed in at least one of the pairwise strain comparisons and 195 clusters differentially expressed in at least three of the seven pairwise comparisons (six pairwise comparisons between testis samples from four strains and one pairwise comparison between spermatogonia samples from two strains) (Datasets EV9 and 10). Analysis

of the data revealed predicted strain-specific mono-directional (e.g., Fig. 2E) and bi-directional (e.g., Fig. 2F) piRNA clusters expressed in testes, as well as strain-specific clusters expressed only in spermatogonia (e.g., Fig. 2G,H). Thus, genomic loci exist within the mouse genome that produce significantly different amount of piRNAs in the germline of different strains of mice.

## Association between an intronic IAP insertion and piRNA production from the mouse protein-coding gene *Nocturnin*

A locus with a notable difference in piRNAs between the four strains was the one overlapping the protein-coding gene *Nocturnin* (*Noct*). *Noct* is one of the 114 previously annotated protein-coding genes that produce piRNAs in the mouse genome (Li et al, 2013). Considering small RNAs mapping uniquely to this locus, *Noct* produces a substantial number of piRNAs in testis samples of only two of the four strains (Fig. 3A). We wondered what may be causing the apparent switch in production of piRNAs from this locus. Because transposons are tightly linked to the function and biogenesis of piRNAs, we turned our attention to known transposon insertions and deletions in the genomes of these strains (Nellåker et al, 2012). In the reference mouse strain, the first intron of *Noct* contains a 5.3 kb ERV insertion of the IAP-IΔ1 subtype and a small fragment of a LINE1 transposon that is variable between the five inbred strains (Fig. 3A). The *Noct* IAP is a recent transposon insertion found in a subset of the laboratory inbred mouse strains (Dupressoir et al, 1999). Laboratory mouse strains have a common origin that dates back to the 1920s, making this insertion potentially less than a century old. Interestingly, mice of the two inbred strains that produce piRNAs from this locus (BL6 and NOD) carry the IAP insertion, while mice of the three strains that produce significantly fewer piRNAs (C3H, 129 and CAST) do not carry the insertion (Fig. 3A). In contrast, production of piRNAs from the locus does not correlate with the presence of the variable LINE1 fragment, since CAST produces very few piRNAs but does not have the LINE1 fragment deletion (Fig. 3A). The perfect correlation between the IAP insertion and pi-Noct abundance raises the possibility of a mechanistic link between IAP insertions and piRNA production.

Because inbred strains differ by many additional variants that are inherited together with the *Noct* IAP insertion and confound the association with piRNA abundance at this locus, we decided to analyze the expression of this piRNA cluster in mice from the outbred strain ICR. In agreement with the results from the inbred strains, we found pi-Noct among the clusters with the highest variation in piRNA abundance (Fig. 3A,B; Dataset EV7; Appendix Fig. S9). Furthermore, 81.4% of pi-Noct expression variation is due

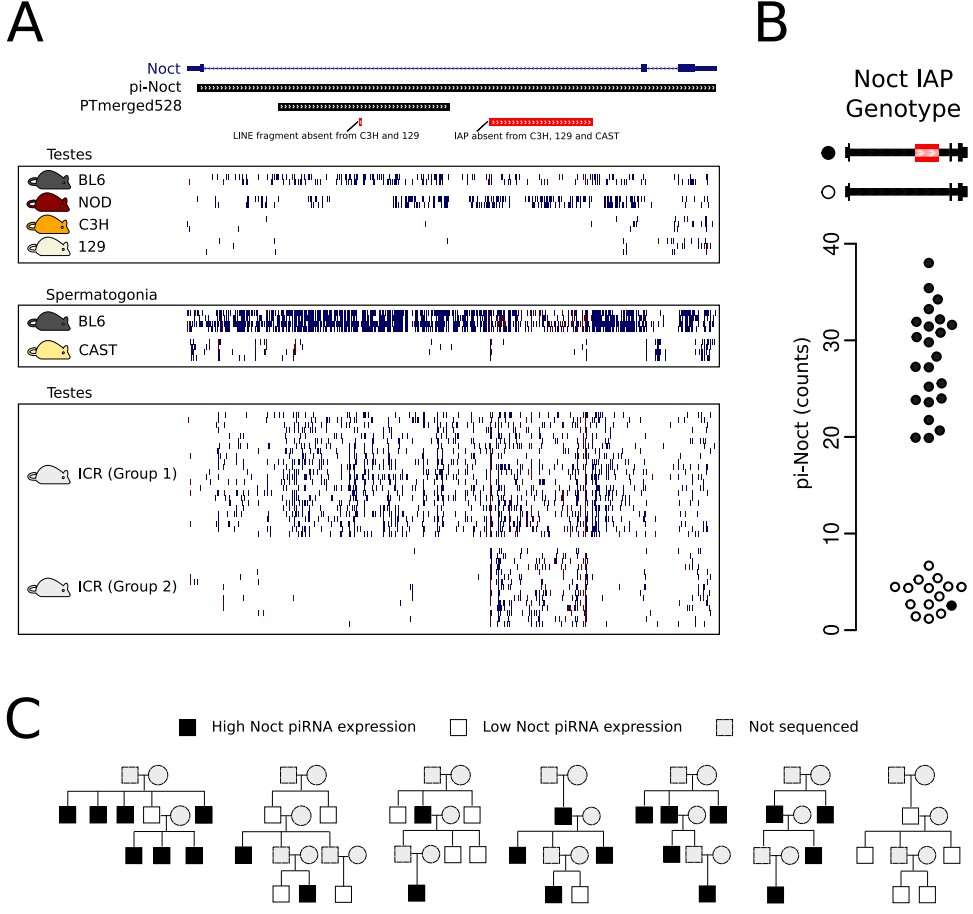

**Figure 3. pi-Noct contains a polymorphic IAP that correlates perfectly with piRNA expression.**

(A) Mouse strains BL6, NOD produce many piRNAs mapping in the sense strand of the gene *Noct*, including its intron. Strains C3H, 129 and CAST produce negligible levels of small RNAs from the same genomic region. Nine samples from the outbred mouse strain ICR contain high levels of pi-Noct small RNAs (group 1), while another nine samples contain low levels (group 2). The eighteen ICR mice were genotyped, as shown in (B). Polymorphic transposable elements from (Nellåker et al, 2012) are indicated. In the schematic at the top, coding exons are indicated as wide blue rectangles, UTRs are thinner blue rectangles, and introns as the lines connecting them. piRNA clusters are indicated as gray rectangles and polymorphic repeats as red rectangles. Small RNAs mapping to the same strand as the gene are shown as blue lines, and those mapping to the antisense strand as red lines. (B) The data points show the normalized counts of small RNAs mapping to the pi-Noct locus in samples from 39 ICR mice. The data points fall into two groups according to their pi-Noct counts. ICR mice for which small RNA data are shown in (B) were genotyped by PCR to test whether they contained the IAP. Samples with low pi-Noct expression were from mice without the IAP (shown as empty circles), while samples with high pi-Noct expression were from mice with at least one copy of the IAP (shown as filled black circles). (C) Closely related mice of the ICR strain have similar pi-Noct expression. Animals with high pi-Noct expression are indicated in black. Only males from whom small RNAs were sequenced and their parents are shown. Animals for which small RNA data is missing are indicated in gray. Squares represent sires and circles represent dams.

to genetics, based on the analysis of the effect of genetic relatedness on piRNA cluster expression in mouse pedigrees of the outbred strain ICR (Fig. 3C; Dataset EV8). To test the link between the variation in piRNA production from this locus and the presence of the IAP in *Noct*, we genotyped these mice and confirmed the association between piRNA production and the IAP insertion (Fig. 3B; Appendix Fig. S10 and "Methods").

In summary, we found that genetic variation is linked to piRNA cluster expression in mice. One of the loci with high piRNA abundance variation between animals overlaps the protein-coding gene *Noct*. The abundance of piRNAs produced from this locus in different animals perfectly agrees with the presence of the IAP in the first intron of the gene. These results suggest that the recent insertion of an IAP at this locus is mechanistically linked to piRNA production.

## General association between piRNA expression and transposable element variants

How pervasive is the association between new transposon insertions or deletions and variation in piRNA production? Although transposons are depleted from genes (Nellåker et al, 2012) as well as from piRNA clusters (Aravin et al, 2006; Girard et al, 2006; Lau et al, 2006), some transposable element variants do overlap the predicted piRNA clusters. We used these annotated transposable element variants to test the association with piRNA cluster expression variation between mouse strains. Indeed, we found that clusters with significant differences in piRNA abundance between mouse strains are more common among clusters with transposable element variants than among the rest of the clusters (Fig. 4A; Dataset EV11). Of three major types of

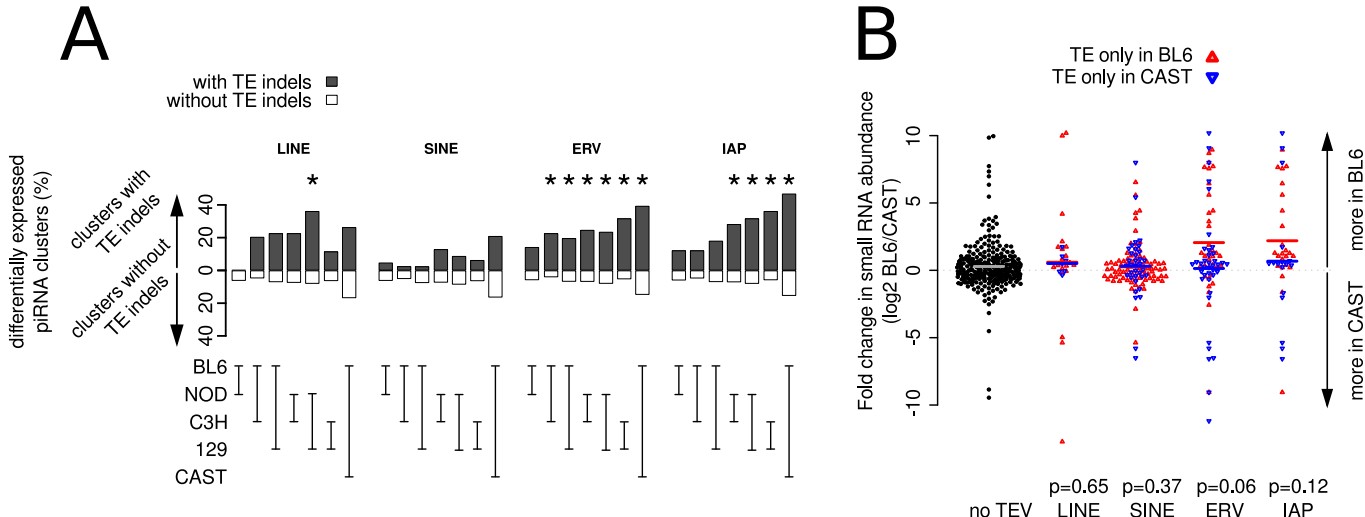

**Figure 4.    Polymorphic ERV insertions are significantly associated with highly variable piRNA production between mouse strains.**

(**A**) Percentages of predicted piRNA clusters with polymorphic transposable elements that are significantly differentially expressed in five strains are shown in black bars. Percentages of predicted piRNA clusters without polymorphic transposable elements are shown in white bars. Significant differences in these percentages, suggesting an association between differential expression and polymorphic transposable elements, were calculated using Fisher's exact test and are indicated with asterisks (*P* < 0.05). The exact *P* values are shown in Table EV1. *N* = 749 predicted piRNA clusters. (**B**) Predicted piRNA clusters with polymorphic ERVs have higher small RNA counts in the strain with the insertion. Polymorphic transposable elements were retrieved from (Nellåker et al, 2012). Data points showing fold change in expression of clusters with insertions only in BL6 are indicated as upward-facing, red triangles. Data points showing fold change in expression of clusters with insertions only in CAST are indicated as downward-facing, blue triangles. Data points showing fold change in expression of clusters without polymorphic transposable elements (no TEV) are shown as filled, black circles. Colored lines indicate group means. For each transposon class, we tested that the distribution of changes in piRNA expression between BL6 and CAST is the same for clusters with transposable elements that are only found in BL6 (red points) as for clusters with transposable elements only in CAST (blue data points), using the Wilcoxon rank-sum test in R (*P* values shown). *N* = 341 (no TEVs), 21 (LINEs in BL6), 7 (LINEs in CAST), 86 (SINEs in BL6), 35 (SINEs in CAST), 35 (ERVs in BL6), 39 (ERVs in CAST), 24 (IAPs in BL6). 17 (IAPs in CAST).

transposons (LINEs, SINEs, and ERVs), we found that the transposons with significant overrepresentation among predicted piRNA clusters with significant differences in piRNA expression between the strains were almost exclusively polymorphic ERVs, especially IAPs (Fig. 4A). Clusters overlapping polymorphic IAPs are few, yet they include some of the piRNA clusters with the biggest differences in piRNA abundance between the mouse strains of this study (such as pi-Noct, 10-qC1-2617, pi-Phf20).

We tested whether transposable element insertions are associated with an increase in piRNA abundance, as seen for pi-Noct. We focused on the comparison between BL6 and CAST because this strain pair has both the highest number of different ERV variants and the highest number of differentially expressed predicted piRNA clusters (Fig. 2D). We split the 195 predicted clusters with variable ERVs between these strains into those with the transposable element only in BL6 (Fig. 4B, data points shown as red triangles pointing up) and those with the transposable element only in CAST (Fig. 4B, data points shown as blue triangles pointing down). We found that the abundance of piRNAs from clusters with ERVs in BL6 tended to be higher in BL6 than in CAST and vice versa (Wilcoxon rank-sum test *P* value = 0.06). The same trend can be seen for clusters with IAP variants but without passing the significance threshold (Wilcoxon rank-sum test *P* value = 0.12). We did not find such a trend between clusters with insertions or deletions of LINEs or SINEs and the direction of piRNA abundance change between the two strains (Fig. 4B). These observations are not due to the expression of the repetitive element itself or an artefact of ambiguous mapping of small RNA data to the mouse

genome, since the changes in small RNA abundance that we report here are based on expression values calculated only from uniquely mapping small RNA reads that align outside annotated repeats. Taken together, the data suggest that ERV insertions can cause an increase of piRNA production or expression from diverse genomic loci in the mouse. ERV insertions could trigger the emergence of novel piRNA clusters during evolution.

## The IAP insertion in *Noct* is associated with post-transcriptional processing of germline-expressed transcripts into piRNAs

IAPs can affect gene expression in multiple ways, one of which is by regulating transcription. Thus, we asked whether piRNA production is explained by IAP-induced ectopic transcriptional activation of the gene during spermatogenesis. *Noct* is a gene known to be expressed in many mouse organs, including testes (Dupressoir et al, 1999). Still, the relative expression of the different *Noct* alleles during spermatogenesis had not been studied. To address this, we analyzed available steady state gene expression data from various stages of spermatogenesis from 129/DBA hybrid mice (Gan et al, 2013) carrying one *Noct* allele with the IAP insertion (inherited from the DBA parent) and the second allele without (inherited from the 129 parent). Using single-nucleotide polymorphisms specific to each of the parental strains, we quantified the expression of the two different alleles in 129/DBA mouse male germ cells and tested whether the *Noct* allele carrying the IAP insertion is more abundantly expressed than the other allele. We found that

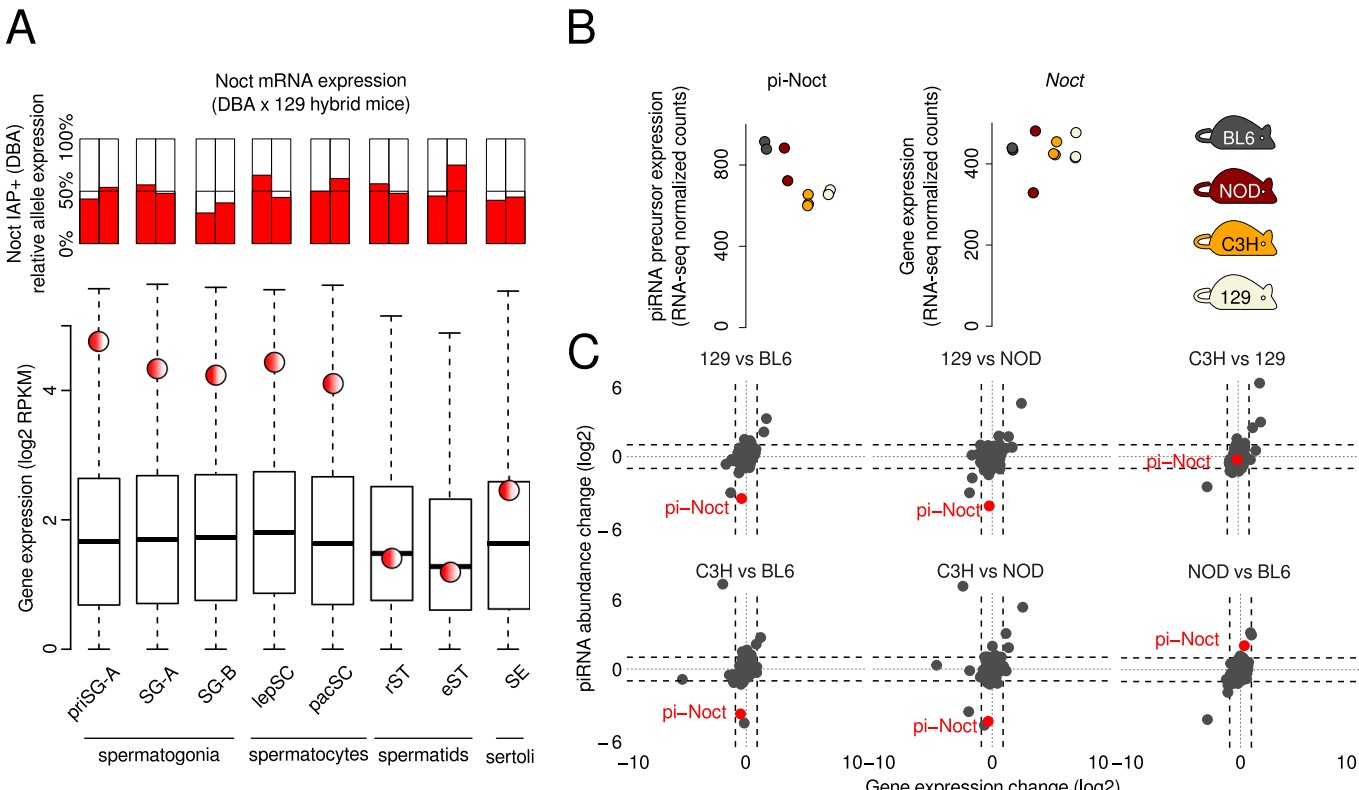

**Figure 5. No evidence of differential mRNA expression due to the presence of the IAP insertion in *Noct*.**

(A) The bar plots show the relative expression of two different *Noct* alleles in 129xDBA F1 hybrid mice carrying one allele with the IAP (inherited from the DBA father) and one without (inherited from the 129 mother). The side-by-side pairs of bar plots correspond to two biological replicates. The y axis shows the percentage of strain-specific RNA-seq reads that map to the DBA *Noct* allele (IAP+). The box plots show the distribution of expression of all genes during spermatogenesis in 129xDBA F1 hybrid mice. The number of genes in each boxplot is 19,796. The red and white circle shows the expression of *Noct* in each cell type. The box plots show the interquartile range, the middle line corresponds to the median, and the whiskers extend to the extreme values provided they are no more than 1.5 times the interquartile range from the box. (B) *Noct* expression in whole testes of the four inbred mouse strains, counting RNA-seq reads on the entire locus, including introns (left panel), and counting reads only on exons (right panel). (C) Comparison of fold change in expression (total RNA-seq) versus fold change in small RNA abundance in pairwise comparisons of the four inbred mouse strains. Points represent the known piRNA-producing loci from (Li et al, 2013).

throughout spermatogenesis, *Noct* is very highly expressed (Fig. 5A, lower panel) with no evidence of the *Noct* allele with the IAP being more highly expressed than the *Noct* allele without the IAP (Fig. 5A, upper panel). Similarly, we analyzed the chromatin state of *Noct* using available H3K4me3 ChIP-seq data from spermatocytes of mouse BL6/CAST hybrid mice (Baker et al, 2015) carrying one *Noct* allele with the IAP insertion (BL6) and another without (CAST). Again, we found that the only active promoter region along the gene is that of the actual *Noct* promoter, with no evidence of additional H3K4me3-marked regions surrounding the IAP, and with the two alleles showing no differences in terms of this active chromatin mark (Appendix Fig. S11). We also sequenced and analyzed *Noct* expression in the testes of the four inbred strains. None of the comparisons revealed a significant difference in *Noct* steady states (Fig. 5B,C; Datasets EV12–16). These results argue that the IAP inserted into an existing germline-expressed gene during very recent murine evolution and that this insertion did not discernibly alter transcription.

An alternative mechanism that could explain the observed data is that the IAP carries a signal involved in post-transcriptional regulation that induces piRNA production from transcripts. This signal is not just

sequence complementarity between an antisense piRNA matching the IAP inside the *Noct* primary transcript, because this would trigger piRNA production only downstream of the IAP. As shown in Figs. 1F and 3A, at this locus, piRNAs are also produced upstream of the IAP, most likely from an intron-retaining isoform (Appendix Fig. S12) transcribed from the first *Noct* promoter. Thus, in this case, it looks like the IAP is causing the unspliced transcript to be exported from the nucleus and to be recognized as a piRNA precursor. We tested whether the association between polymorphic IAP insertions and piRNA production depends on the orientation of the IAP. Comparing small RNA production from predicted piRNA clusters from BL6 and CAST spermatogonia, we found no difference in piRNA levels at loci with strain-specific ERVs antisense to the piRNA cluster (Fig. 6A, lower panel, Fig. 6B, right panel). Importantly, however, strong and significant associations were observed specifically where ERV insertions are in the piRNA-producing strand (Fig. 6A, upper panel, Fig. 6B, left panel). Thus, we conclude that ERV insertions can trigger and/or enhance piRNA production from existing transcribed genomic loci, likely through a post-transcriptional mechanism and that this mechanism appears to require the ERV to be oriented in sense to the host transcript.

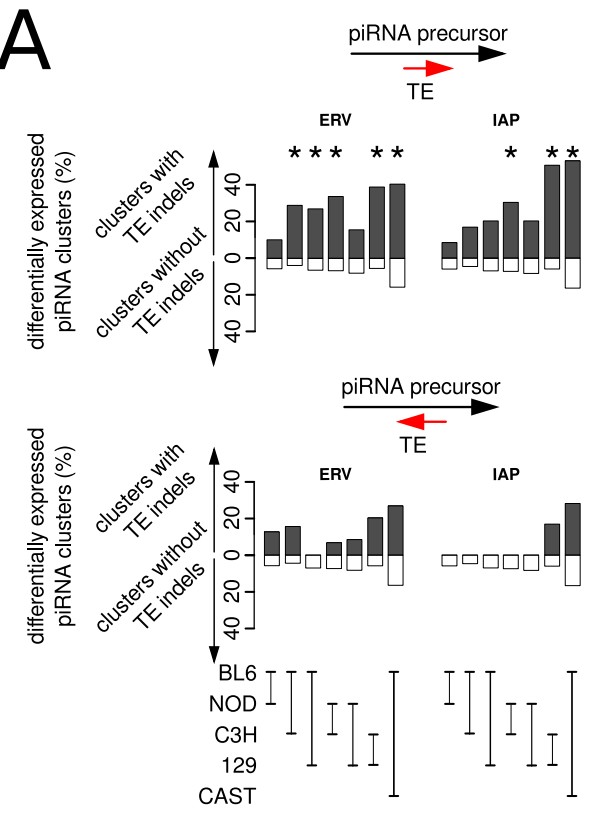

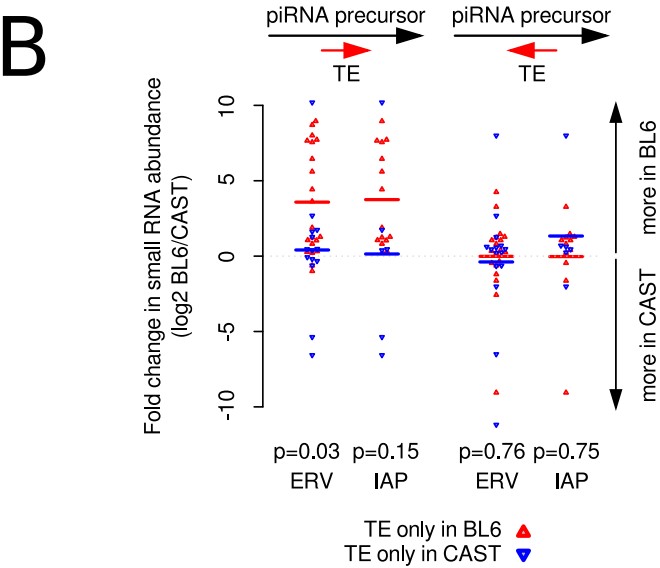

Figure 6.   ERV insertions specifically on the sense strand of the precursor transcript are associated with higher piRNA production between mouse strains.

(A) Predicted piRNA clusters with ERV insertions in the sense strand (upper panels) and in the antisense strand (lower panels) were analyzed separately. Percentages of predicted piRNA clusters with polymorphic transposable elements that are significantly differentially expressed in five strains are shown in black bars. Percentages of predicted piRNA clusters without polymorphic transposable elements are shown in white bars. Significant differences in these percentages, suggesting an association between differential expression and polymorphic transposable elements, were calculated using Fisher's exact test and are indicated with asterisks ($P < 0.05$). The exact $P$ values for the upper panel are shown in Table EV2 and those for the lower panel are shown in Table EV3. $N = 749$ predicted piRNA clusters. (B) Predicted piRNA clusters with polymorphic ERVs have higher small RNA counts in the strain with the insertion, only when the insertion is in the same strand as the predicted cluster. Polymorphic transposable elements were retrieved from (Nellåker et al, 2012). Data points showing fold change in expression of clusters with insertions only in BL6 are indicated as upward-facing, red triangles. Data points showing fold change in expression of clusters with insertions only in CAST are indicated as downward-facing, blue triangles. For each transposon class, we tested that the distribution of changes in piRNA expression between BL6 and CAST is the same for clusters with transposable elements that are only found in BL6 (red points) as for clusters with transposable elements only in CAST (blue data points), using the Wilcoxon rank-sum test in R ($P$ values shown). $N = 29$ (ERV on sense strand in BL6), 19 (IAP on sense strand in BL6), 10 (ERV on sense strand in CAST), 4 (IAP on sense strand in CAST), 21 (ERV on antisense strand in BL6), 14 (IAP on antisense strand on BL6), 12 (ERV on antisense strand in CAST), 6 (IAP on antisense strand in CAST).

different strains of flies (Song et al, 2014). In addition to quantitative differences in piRNA production, both mice and flies have strain-specific sources of piRNAs. Therefore, the rapid emergence of multiple new piRNA-producing loci within a species is a core property of the piRNA system and likely to be found in all animals expressing this pathway. The high within-species diversity also agrees with the high between-species divergence of piRNA-producing loci in animals (Assis and Kondrashov, 2009; Chirn et al, 2015).

One of the primary mechanisms of novel piRNA production in a species appears to be the insertion of transposable elements to new positions in the genome. In mice, we found a significant association between piRNA production and novel transposable element insertions, in particular IAPs. In flies, strain-specific piRNA clusters were found at positions of novel insertions of LTR and LINE elements (Mohn et al, 2014; Shpiz et al, 2014; Song et al, 2014). Our analysis revealed that transposable element insertions or deletions are often—but not always—found at clusters that show major inter-strain differences in piRNA production. There are several possible explanations for this. It is possible that the annotation of transposable element variants is incomplete, and all strain-specific piRNA-producing loci are due to transposable elements insertions and deletions. In addition, some of the differences in piRNA abundance could be due to differences in the expression level or the processing of the precursor. For example, genetic variation leading to the gain of a new binding site for A-Myb, the major transcription factor for pachytene piRNA expression, could explain the birth of some of the strain-specific piRNA clusters. Differences in the steady state level of the precursor was not the case for one of the highly variable piRNA clusters that we studied in detail (pi-Noct), but it may be the case for others.

## Discussion

We uncovered significant variation in piRNA production from a subset of piRNA-producing loci in genetically diverse mouse strains. In particular, for some piRNA-producing loci we found that genetic differences explain most of the variation in expression between different mice. This is the first comparison of piRNA production in different animals of any mammalian species. Our results are in agreement with what was previously observed in

We found that the orientation of the ERV insertion appears coupled to inter-strain piRNA cluster expression differences. In particular, we found that ERV insertions only had an effect when they were in sense to the piRNA precursor. The IAP insertion in the first intron of the gene *Noct* fits this model. In the case of pi-Noct, we found that the IAP insertion does not modify the expression level of the precursor. It is also clear that piRNA production from *Noct* by antisense piRNAs targeting the IAP is not the mechanism that leads to piRNA production from this locus. This is because most piRNAs are produced from the intron, upstream of the position of the IAP insertion. The fact that piRNAs are produced from the intron, also highlights that *Noct* transcripts producing piRNAs are either unspliced or aberrantly spliced, that they evade surveillance mechanisms in the nucleus and that they are exported to the cytoplasm, where piRNA biogenesis takes place. Moreover, that IAP-containing, unspliced *Noct* transcripts produce piRNAs suggests that these transcripts are processed into piRNAs because they are recognized as retroviruses, like KoRV-A in Koala and the AKV Murine Leukemia Virus in the AKR mouse strain (Yu et al, 2019). However, unlike KoRV-A and AKV elements, the IAP is embedded within the intron of a gene and transcribed with it. It reveals how an ERV insertion in the intron of a protein-coding gene can signal a much greater transcript for piRNA biogenesis. We can only speculate about the mechanism of piRNA production from the IAP-containing allele of *Noct*. It could be the absence of splicing, as previously proposed by Yu et al, that signals this transcript should be sliced into piRNAs. There is the association between intron retention (and unusually long first exons) and piRNA production in both mammals and flies (Yu et al, 2019, 2021; Zhang et al, 2014). It could also be the presence of a strand-specific signal within the IAP interacting with a nuclear exporter and piRNA biogenesis factor (Zolotukhin et al, 2008). Functional experiments are necessary to further dissect the mechanism by which the *Noct* IAP insertion leads to piRNA production from this locus.

We currently do not know whether the differences in piRNA content between animals of the same strain have biological or physiological consequences, conferring higher or lower fitness. We speculate that the burst of novel piRNAs triggered by a transposon insertion event has the potential to generate new regulatory effects in *cis* and in *trans*. As with other genetic variants, the emergence of new piRNAs can be beneficial (recognition of invading parasitic elements) or deleterious (silencing of essential protein-coding genes) for the organism. What is unique to polymorphic piRNA-producing loci is the magnitude of new material for natural selection to act upon. It is perhaps because of the many possibilities for positive or negative effects on fitness by each individual piRNA, among the many produced from a single locus, that piRNA-producing loci are gained and lost so fast during evolution.

In conclusion, by sequencing and analyzing small RNAs from male germlines of different mouse strains, we identified polymorphic and variably expressed piRNA-producing loci in a mammalian species. Insertions and/or deletions of active ERVs at germline-expressed genomic loci are two genetic mechanisms that spark piRNA expression variation and diversity, but there are certainly more to be discovered. Here, we analyzed piRNAs in a handful of mouse strains but there are many more mouse strains with sequenced genomes that could be used to study piRNA expression variation in this mammal. Although small RNA data

from inbred mouse strains were essential for the documentation of within-species differences in piRNAs determined by genetics, they could not be used to identify genetic variants associated with piRNA cluster expression in a global, systematic and unbiased way, because classical inbred strains vary at millions of positions along their genomes (Keane et al, 2011). Nevertheless, the data we generated chart the degree of piRNA cluster expression variation to be expected between genetically different animals of this rodent species. Some mouse strains have very young copies of IAPs consistent with activity in recent years (Nellåker et al, 2012). It remains to be seen whether ERV insertions and deletions are also a significant source of piRNA expression variation and diversification in other mammalian species, such as human.

# Methods

### Reagents and tools table

| Reagent/resource | Reference or source | Identifier or catalog number |
|---|---|---|
| **Experimental models** | | |
| 129S1/SvlmJ (*Mus musculus*) | Colony established from animals purchased from The Jackson Laboratory | |
| NOD (*Mus musculus*) | Colony maintained at the animal facility of the Germans Trias i Pujol Research Institute | |
| C3HeB/FeJ (*Mus musculus*) | Colony maintained at the animal facility of the Germans Trias i Pujol Research Institute | |
| C57BL/6J (*Mus musculus*) | The Jackson Laboratory | |
| CAST/EiJ (*Mus musculus*) | The Jackson Laboratory | |
| **Antibodies** | | |
| CD9 | eBioscience | 17-0091-82 |
| Epcam | eBiocience | |
| **Chemicals, enzymes, and other reagents** | | |
| Collagenase IV | Worthington | LS004189 |
| DMEM | Gibco | 31966-024 |
| Penicillin–streptomycin | Gibco | 15140-122 |
| Fungizone | Gibco | 15290-018 |
| Gentamycin | Serva | 4799.01 |
| EDTA | Sigma | T4849 |
| PBS | Gibco | 14190-094 |
| SUPERase.in | Invitrogen | AM2696 |
| TRIzol | Invitrogen | 1596018 |
| DNAse | Invitrogen | AM1906 |
| DNA purification kit | Promega | |
| Phusion High Fidelity DNA Polymerase | Life Technologies | |
| SYBR safe | Life Technologies | |

| Reagent/resource | Reference or source | Identifier or catalog number |
|---|---|---|
| **Software** | | |
| Bedtools v2.30.0 | Quinlan and Hall, 2010 | |
| Bowtie v1.3.1 | Langmead et al, 2009 | |
| Bowtie2 v2.2.5 | Langmead and Salzberg 2012 | |
| Cutadapt v2.10 | Martin 2011 | |
| DESeq2 v1.34.0 (R package) | Love et al, 2014 | |
| EMBOSS Stretcher v6.5.7.0 | Myers and Miller, 1988; Madeira et al, 2024 | |
| ENSEMBL Compara Perl API (ENSEMBL release 105) | Cunningham et al, 2022 | |
| FASTX Toolkit v0.0.14 | https://github.com/agordon/fastx_toolkit | |
| featureCounts v2.0.1 | Liao et al, 2014 | |
| HISAT2 v2.2.1 | Kim et al, 2019 | |
| liftOver | Hinrichs et al, 2006 | |
| lme4 v1.1-35.5 (R package) | Bates et al, 2015 | |
| Lme4breeding v1.0.50 (R package) | Covarrubias-Pazaran, 2024 (preprint) | |
| pedigreemm v0.3-5 (R package) | Vazquez et al, 2010 | |
| proTRAC tools (NGS Toolkit) v2.1 | Rosenkranz et al, 2015 | |
| proTRAC v2.4.4 | Rosenkranz and Zischler, 2012 | |
| R v4.2.3 | R Core Team, 2023 | |
| Samtools v1.10 | Li et al, 2013 | |
| SNPSplit v0.3.3 | Krueger and Andrews, 2016 | |
| sva v3.54.0 (R package) | Leek et al, 2012 | |
| **Other** | | |
| Strainer | BD Falcon | 352340 |
| Molecular Imager Gel® Doc™ XR+ imaging system | Biorad | |

## Mouse tissue isolation and RNA extraction

Testes used in this study were obtained from mice from various sources, all following institutional regulations for animal care and use. Specifically, ICR (ICR-CD1, Envigo) and 129S1/SvlmJ (local established colony from previously purchased animals from The Jackson Laboratory) mice were maintained and used according to the guidelines of the Universitat de Barcelona Animal Care and Use Committee, C3HeB/FeJ and NOD mice were maintained and used according to the guidelines of the animal facility of the Institute Germans Trias I Pujol research institute (IGTP). All testes used in this work were from young adult mice. Testes were rapidly dissected, snap-frozen in liquid nitrogen and stored at −80 °C. For the sequencing of small RNA from classical inbred mouse strains, total RNA was extracted from previously frozen testes using TRIzol

Reagent (Life Technologies; Thermo Fisher Scientific) linked to PureLink RNA Mini Kit (Invitrogen: Thermo Fisher Scientific) following the manufacturer's protocol "TRIzol Plus Total Transcriptome Isolation".

For the isolation of spermatogonial RNA, C57BL/6J and CAST/EiJ mice were obtained from The Jackson Laboratory and kept in the SPF animal facility of Max Planck Institute of Immunobiology and Epigenetics until sacrifice. In order to isolate spermatogonia from mice, testes were dissected and digested according to the protocol by Liao et al (Liao et al, 2016), with minor modifications. Briefly, we euthanized 6-week-old mice with $CO_2$ and quickly dissected the testes, removed the tunica albuginea and loosened the seminiferous tubules. We then digested these tissues with 1 mg/ml collagenase IV (Worthington, LS004189) in DMEM (Gibco, 31966-024) supplemented with 10% FBS, 100 U/ml penicillin–streptomycin (Gibco, 15140-122), 250 ng/ml fungizone (Gibco, 15290-018) and 50 µg/ml gentamycin (Serva, 4799.01) in a petri dish at 37 °C over a Thermoblock, shaking at 600 rpm for 30 min. The reaction continued for another 10 min after the addition of 0.25% trypsin EDTA (Sigma, T4849) at 37 °C and 600 rpm. We homogenized the digested tissues by pipetting up and we washed the solution with a double amount of PBS (Gibco, 14190-094) supplemented with 10% FBS. Pieces of remaining, undigested tissues were filtered with a 40-µm strainer (BD Falcon, 352340). The filtered solution was then centrifuged at $300 \times g$ for 10 min at 4 °C. We removed the supernatant and then resuspended the pellet in 200 µl of FACS buffer (PBS supplemented with 5% BSA and 5 mM EDTA) supplemented with 1U/µl SUPERase.in (Invitrogen, AM2696). Spermatogonia were sorted according to (Kanatsu-Shinohara et al, 2011) for the expression of CD9 (eBioscience, 17-0091-82, 1 µg) and Epcam (eBioscience, 0.125 µg). Sorted cells were centrifuged at $300 \times g$ for 10 min at 4 °C and resuspended in 1 ml of TRIzol (Invitrogen, 15596018). Spermatogonial RNA was purified according to the standard TRIzol protocol, and contaminant genomic DNA was digested using the DNA-free kit (Invitrogen, AM1906).

## RNA-seq library preparation and sequencing

Small RNA sequencing libraries were prepared by the Genomics and Bioinformatics Facility of the IGTP. Libraries were prepared with TruSeq small RNA from illumina with extended range of size selection. Pippin prep was used for automated pooled library size selection. Libraries were indexed using Illumina barcodes and sequenced using a HiSeq2500 (Illumina) as single 50 nt reads. Small RNA libraries corresponding to samples from inbred strains were sequenced as a single pool on two lanes and the resulting data (all showing very high correlation between lanes) were merged for analysis. Long RNA was sequenced using TruSeq Stranded Total RNA with Ribo-Zero H/M/R_Gold libraries, 150 bp paired-end, by Macrogen using a NovaSeq6000. All RNA samples passed quality controls ($RIN \geq 7$ and rRNA ratio $\geq 1$) before library preparation. It should be noted that sequence differences between the strains may also affect the adapter ligation and PCR amplification steps of the small RNA sequencing protocols, leading to distortion in the abundance of individual piRNAs between strains. However, this limitation is unlikely to affect differential expression of piRNA clusters because the latter usually consist of hundreds of different piRNAs, most of which have the same sequence between strains. Also, reads mapping to repeats, polymorphic or fixed, were not used for the analysis of differential piRNA expression.

## Small RNA-seq data analysis

We removed the adapter (TGGAATTCTCGGGTGCCAAG-GAACTCCAGTCAC) from the small RNA reads using cutadapt v2.10 (Martin, 2011), requiring 9 nt of match with the adapter. We discarded reads shorter than 19nt, longer than 36 nt and any reads not matching the adapter. We filtered reads based on quality using the FASTX Toolkit v0.0.14 (https://github.com/agordon/fastx_toolkit) allowing a minimum quality score 30 over at least 90% of nucleotides. We mapped the trimmed and filtered reads to the genome using bowtie v1.2.3 (Langmead et al, 2009) with the options –M 1 --best --strata -v 1 to get the best alignment with up to 1 mismatch, reporting only one random match for multi-mapping reads. Reads were mapped to the most relevant available sequenced mouse strain genome; for samples from C57BL/6J mice we used the primary assembly of the reference mouse genome (GRCm38/mm10), for samples from CAST/EiJ we used the CAST_EiJ_v1 assembly, for samples from 129S1/SvlmJ we used the 129S1_SvImJ_v1 assembly, for samples from C3HeB/FeJ we used the C3H_HeJ_v1 assembly, for samples from NOD mice we used the NOD_ShiLtJ_v1 assembly and for samples from the ICR outbred mouse strains we used the GRCm38/mm10 primary assembly. Coordinates of different strains were converted to the reference mouse genome mm10 assembly using ENSEMBL Compara Perl API (ENSEMBL release 105) (Cunningham et al, 2022) using the EPO murinae multiple alignments. The coordinates of the piRNA clusters from (Li et al, 2013) in other strain genomes are provided in Dataset EV17. The proportions of reads mapping uniquely at Li et al 2013 clusters, are provided in Dataset EV18.

## Prediction of piRNA-producing loci

We used the proTRAC pipeline v2.4.4 (Rosenkranz and Zischler, 2012) to predict clusters for each of the 18 samples from inbred mice with default options. Briefly, using proTRAC tools (Rosenkranz et al, 2015), we collapsed and removed the low-complexity reads and mapped the remaining to the respective genome. Multimapping reads were reallocated based on the mapping rate in the upstream and downstream 10Kb flanking sequence, using a 1-Kb sliding window. using a bell-shaped function and reporting only loci with allocated reads. The resulting reallocated read file was used to predict piRNA clusters in each strain genome, using proTRAC with default parameters. Clusters were annotated as mono-directional or bi-directional by proTRAC, depending on whether at least 10% of the reads mapping to the locus were on the antisense strand. As each mouse strain has a different coordinate system, we converted all coordinates to those of the mouse reference genome using the murinae multiple alignment in ENSEMBL Compara (ENSEMBL release 105). Note that conversion of coordinates between strains, using ENSEMBL Compara and liftOver (Hinrichs et al, 2006) (for conversion between mm39 and mm10) lead to the loss of some clusters due to duplications or splitting of genomic regions in different strain genome assemblies. The clusters that did not convert one-to-one between strains were excluded from downstream analysis. For each strain, piRNA clusters were defined as the union of the piRNA clusters predicted from each sample, calculating the union of overlapping genomic intervals, only merging clusters with the same strand assignment by proTRAC. From this set, we removed clusters that matched repeats

(RepeatMasker annotation) or polymorphic transposable elements (Nellåker et al, 2012) by more than 80% of their length. The remaining clusters were then converted back to each of the strain genome coordinates using the murinae multiple alignment in Ensembl Compara, resulting in 852 clusters present in the five strains. This set included some pairs of predicted clusters overlapping each other where one cluster was annotated as "bi-directional" and the other as "mono-directional". To avoid double-counting, we removed clusters predicted to be bi-directional that overlapped mono-directional clusters. We also removed clusters with total read counts of less than 10 in all the samples. The final set consists of 749 mouse-predicted piRNA clusters. The coordinates of these regions in the mouse reference genome and the results of the differential expression analysis in testis of four inbred strains and in spermatogonia of two inbred strains are provided in Datasets EV9 and EV10, respectively. The coordinates of the predicted clusters in strain genomes are provided in Dataset EV19. The percentages of reads mapping uniquely at the predicted piRNA-producing loci are provided in Dataset EV20.

## Differential expression analysis of piRNA-producing loci and test of association with variable transposable elements

Differential expression was analyzed only for regions that were in alignment in all five inbred strains. Of the 214 piRNA clusters from Li et al (Li et al, 2013), 192 were in genomic regions aligned among all strains. To quantify the expression of piRNA clusters (known or predicted), we annotated reads mapping to the clusters using featureCounts v2.0.1 (Liao et al, 2014) with the options –Q 1 -s 0 –minOverlap 18 to count reads with minimum quality score of 1, mapping within the region of the cluster with a minimum overlap of 18nt. To reduce possible artefacts due to differences in repeat content or repeat expression between strains, for differential expression analysis, we only counted reads mapping to unique locations in the genome. The same analyses, also removing reads overlapping repeats from the RepeatMasker annotation, gave similar results (Datasets EV21–EV23). We removed from differential expression analysis clusters with fewer than ten reads in all samples. Predicted clusters overlapping (for example, in cases where a mono-directional and a bi-directional cluster overlapped) were removed from statistical tests. We used DESeq2 v1.34.0 (Love et al, 2014) to normalize read counts with respect to library size. To study quantitatively the effect of genetics on piRNA cluster expression using linear models, we used the Variance Stabilizing Transformation (VST) (Anders and Huber, 2010) to transform the piRNA cluster expression values. To assess the effect of strain on piRNA cluster expression *globally*, we used the lme4 package (Bates et al, 2015) and modeled strain as a fixed effect and piRNA cluster as a random effect (expression ~ strain + (1|cluster)). To assess the effect of strain on piRNA cluster expression for each cluster separately, we tested whether strain as a fixed effect explains a significant fraction of inter-individual piRNA cluster expression variation (expression ~ strain). The adjusted R-squared, the nominal *P* value and the *P* value adjusted with the Benjamini-Hochberg correction (Benjamini and Hochberg, 1995) are shown in Dataset EV4. To compare the genetic distance to the expression distance between the four classical inbred strains, for each piRNA cluster separately, we calculated the genetic (euclidean) distance between the four strains using known SNPs (dbSNP142) overlapping the cluster, also calculated

the expression (euclidean) distance using the VST-transformed piRNA cluster expression values and compared the tree of the genetic and expression distances using the Robinson-Foulds metric and the Adjusted Rand Index matric (Dataset EV6). For the estimation of heritability of piRNA cluster expression in the samples of the ICR strain, because the samples were sequenced in two batches, we batch-corrected the library size-adjusted and VST-transformed piRNA cluster expression values using the sva R package (Leek et al, 2012). For the estimation of piRNA cluster expression heritability, we first calculated the Additive Relationship Matrix using pedigreemm package (Vazquez et al, 2010) from the mouse pedigrees (Appendix Fig. S8). To estimate heritability we used the lme4breeding R package (Covarrubias-Pazaran, 2024), where expression was modeled with the individual as a random effect linked to the Additive Relationship Matrix and the pedigree set as a fixed effect (three pedigrees belong to one set and the other four in a different set, where the male ancestors of the second set were overnurished). The variance of piRNA cluster expression due to genetics was also calculated in 100 permuted pedigrees, where animals of the same generation and the same pedigree set were shuffled, maintaining animals in the same generation and the same pedigree set. For differential piRNA cluster expression analysis between inbred mouse strains, we used DESeq2 v1.34.0 (Love et al, 2014) with absolute $\log_2$ fold change threshold greater than 1 and false discovery rate threshold of 0.05. No blinding of sample identifiers was done.

We retrieved the annotation of variable transposable elements from Additional file 13 from Nellaker et al, (Nellåker et al, 2012). We grouped the retrieved variable transposable elements into SINEs (as annotated in the file), LINEs (annotated as LINE and LINE fragments), ERVs (the rest of the elements in the retrieved file, which are different families of ERVs) and IAPs (annotated as IAP-I). For association with predicted piRNA clusters, we considered that predicted clusters overlapped variable transposable elements if they were within 5 kb of one and tested the association using Fisher's exact test in R with significance threshold of 0.05. For tests of association using strand information, we only considered clusters and repeats with annotated strand as + or − (bi-directional piRNA clusters were excluded from this analysis). The significance of the differences in the distribution of fold change expression of predicted clusters in different strains (Fig. 6B) was based on the two-sample Wilcoxon rank-sum test in R.

## RNA-seq and ChIP-Seq allele-specific data analysis

To test for differences in expression of *Noct* IAP+ and *Noct* IAP− alleles, we retrieved RNA-seq data from GSE35005 (Gan et al, 2013) from DBA/2NCrlVr x 129S2/SVPasCrlVr F1 hybrid mice. According to the annotated variable transposable elements of eighteen genotyped mouse strains (Nellåker et al, 2012), three 129 strains (129S1/SvImJ, 129P2/OlaHsd and 129S5/SvEvBrd) carry the IAP insertion in *Noct*. We thus expect that 129S2/SVPasCrlVr also carries it. As noted in Nellaker et al, mouse substrains are nearly identical to each other in comparison to other strains. Similarly, following the same line of thought we expect that the DBA/2NCrlVr strain carries the same *Noct* allele as the genotyped strain DBA/2J. To test for differences in the H3K4me3 chromatin mark on a *Noct* IAP+ and a *Noct* IAP− allele, we retrieved ChIP-seq data from GSE60906 (Baker et al, 2015) from C57BL/6J × CAST/EiJ F1 hybrid mice. According to the genotyped mouse strains, C57BL/6J carries the *Noct* IAP+ allele and CAST/EiJ carries the *Noct* IAP− allele.

To retrieve reads mapping to the two different alleles in the samples of the hybrid mice, we used SNPSplit v0.3.3 (Krueger and Andrews, 2016). Briefly, we masked the reference mouse genome changing all the SNP positions to Ns. The list of SNPs between mouse strains and the reference mouse genome was retrieved from the Sanger Institute Mouse Genomes Project v5, dbSNP142. RNA-seq reads were mapped using HISAT2 v2.2.1 (Kim et al, 2019) with options –no-softclip using known splice sites from the reference mouse genome. ChIP-seq reads were mapped using bowtie2 v2.2.5 (Langmead et al, 2012) with default options. Reads overlapping SNPs were assigned to the corresponding strain using SNPSplit.

## Noct allele genotyping of ICR mice

For genotyping the *Noct* IAP (Fig. 3B; Appendix Fig. S10), DNA extraction from mouse liver tissue was performed using the Maxwell 16 Tissue DNA Purification kit (Promega) following the manufacturer's protocol. In total, 20 μL PCR reactions were performed with 100 ng of genomic DNA using the Phusion High Fidelity DNA Polymerase (2 U/μL) (Life Technologies) following the manufacturer's instructions. Specifically, 0.5 μL of 10 μM Forward primer (5′-AAGAGCTAATCCTGTACATGGC-3′) and 0.5 μL of 10 μM Reverse primer (5′-CCAGGGTAACACAGTGA-GACG-3′) were used together with 4 μL Buffer GC (5×), 0.4 μL 10 mM dNTPs and 0.4 μL of Phusion Polymerase. PCR conditions were as follows: an activation step at 98 °C for 3′; 30 × 3-step cycles of denaturing at 98 °C for 10″, annealing at 63.1 °C for 20″ and extension at 72 °C for 4′ 19″; followed by a final step at 72 °C for 5′. Amplicons were run in 1.5% agarose gels stained with SYBR Safe (Life Technologies). Gel pictures were taken with the Molecular Imager® Gel Doc™ XR+ imaging system (BioRad).

## Cross-species conservation analysis

We converted the 214 piRNA clusters annotated by (Li et al, 2013) in mouse (mm10) to the genomic coordinates systems of human (hg19), macaque (RheMac8), rat (rn6), marmoset (CalJac3), opossum (MonDom5) and platypus (OrnAna1) with liftOver using a minimum match (--minMatch) of 0.1. The loci that were converted one-to-one were considered conserved/syntenic. To get which of the conserved loci retained the ability to produce piRNAs, we overlapped them to piRNA-producing loci annotated in each of the species by (Özata et al, 2020).

## Sequence similarity analysis

To perform the pairwise similarity analysis (Appendix Fig. S6), we extracted the sequence of known piRNA loci (Li et al, 2013), predicted piRNA loci and lncRNA genes with orthologs in all the analyzed strains using bedtools (Quinlan and Hall, 2010) getfasta. We computed pairwise alignments of each locus between all possible combinations of strains using EMBOSS Stretcher (Myers and Miller, 1988; Madeira et al, 2024) that performs a Needleman-Wunsch algorithm. This resulted in a pairwise alignment in FASTA format, that was further processed in R to compute the percentage of identical bases in the pairwise alignment excluding Ns in any of the aligned sequences. Note that only loci without overlap to protein-coding genes were used for this analysis. For the pairwise similarity analysis of differentially expressed and not differentially

expressed pachytene piRNA clusters (Appendix Fig. S7A), we took all pachytene clusters regardless of their overlap to protein-coding genes. We obtained the A-Myb and Tcfl5 ChIP-seq peaks from (Yu et al, 2023), found those peaks that overlapped promoters in mm10, converted them to the rest of the strains using ENSEMBL Compara, and repeated the pairwise similarity analysis of TF-bound promoters for DE and not DE piRNA clusters (Appendix Fig. S7C).

## Splicing efficiency analysis

Intron retention in *Noct* (Appendix Fig. S12) was measured by relative intron expression, the coverage in the intron divided by the mean coverage in the surrounding exons, and splicing efficiency, the ratio of split versus total reads at splice sites. To calculate relative intron expression, introns were first obtained by subtracting the exons to their respective transcripts and RNA-seq coverage on exons and introns was independently estimated with feature-Counts. To calculate splicing efficiency, splice sites were obtained by taking extremes of introns and extending them by 7 bp inside the exon and 7 bp inside the intron. Then, RNA-seq BAM alignments were converted to BED, separating split and unsplit reads by the presence of N in the CIGAR line. Split and unsplit reads mapping to splice sites were counted with bedtools intersect with options -c -f 0.5.

## Data availability

All sequencing data produced for this publication are available from https://www.ncbi.nlm.nih.gov/geo/query/acc.cgi?acc=GSE215030. Custom scripts written to analyze the data are available from https://github.com/vavouri-lab/mouse-piRNA-variation.

The source data of this paper are collected in the following database record: biostudies:S-SCDT-10_1038-S44318-025-00475-4.

## Peer review information

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

## Acknowledgements

The authors thank Pere-Joan Cardona from the Microbiology Service at Hospital Germans Trias i Pujol and Jorge Díaz, from the CMCiB for kindly providing tissues of the C3HeB/FeJ mouse strain, Marta Vives-Pi from the Immunology of Diabetes Unit at Germans Trias i Pujol Research Institute (IGTP) for tissues of the NOD mouse strain and Jordi Llorens from the Department of Physiological Sciences, University of Barcelona for tissues of the 129S1/SvlmJ strain. The authors thank Lauro Sumoy and the High Content Genomics and Bioinformatics Facility of the IGTP for small RNA library preparation, the IGTP and the IJC High Performance Computing Core Facilities for systems administration and the CRG Genomics Unit for small RNA sequencing. The authors thank Ben Lehner, Adelheid Lempradl and Anita Ost for comments on an earlier version of the manuscript. EC was funded by an AGAUR FI PhD fellowship. This work was funded by the Spanish Ministry of Economy and Competitiveness Grant BFU2015-70581 and PID2019-111676GB-I00 to TV and PID2019-105278RB-I00 to SVF. Research at the IJC is supported by the Fundació Internacional Josep Carreras and the CERCA Programme/Generalitat de Catalunya. AMV was supported by a predoctoral fellowship LCF/BQ/DI22/11940007 from the "la Caixa" Foundation (ID 100010434).

## Author contributions

**Eduard Casas**: Conceptualization; Data curation; Formal analysis; Investigation; Visualization; Methodology; Writing—original draft. **Adrià Mitjavila-Ventura**: Conceptualization; Data curation; Formal analysis; Funding acquisition; Investigation; Visualization; Methodology; Writing—original draft; Writing—review and editing. **Pío Sierra**: Formal analysis; Investigation. **Cristina Moreta-Moraleda**: Validation; Investigation. **Judith Cebria**: Investigation. **Ilaria Panzeri**: Resources; Investigation. **J Andrew Pospisilik**: Resources; Funding acquisition. **Josep C Jimenez-Chillaron**: Resources; Supervision; Funding acquisition. **Sonia V Forcales**: Resources; Supervision; Funding acquisition; Validation; Investigation; Methodology; Writing—review and editing. **Tanya Vavouri**: Conceptualization; Resources; Data curation; Formal analysis; Supervision; Funding acquisition; Investigation; Visualization; Methodology; Writing—original draft; Project administration; Writing—review and editing.

Source data underlying figure panels in this paper may have individual authorship assigned. Where available, figure panel/source data authorship is listed in the following database record: biostudies:S-SCDT-10_1038-S44318-025-00475-4.

## Disclosure and competing interests statement

The authors declare no competing interests.

