## [Peer Review File · The EMBO Journal]

Genetic polymorphisms lead to major, locus-specific, variation in piRNA production in mouse.

Eduard Casas, Adrià Mitjavila Ventura, Pío Sierra, Cristina Moreta-Moraleda, Judith Cebria, Ilaria Panzeri, John Pospisilik, Josep Jimenez-Chillaron, Sonia Forcales, and Tanya Vavouri

Corresponding author(s): Tanya Vavouri (tvavouri@carrerasresearch.org) , Sonia Forcales (sforcales@ub.edu)

Review Timeline:

Transfer from Review Commons:	16th Jul 24
Editorial Decision:	25th Sep 24
Revision Received:	23rd Jan 25
Editorial Decision:	19th Feb 25
Revision Received:	23rd Apr 25
Accepted:	13th May 25

Review
COMMONS

Editor: Cornelius Schneider

Transaction Report: This manuscript was transferred to The EMBO JOURNAL following peer review at Review Commons.

Review #1

1. Evidence, reproducibility and clarity:

Evidence, reproducibility and clarity (Required)

This manuscript was a pleasure to read. The function of non-transposon-silencing piRNAs, particularly the pachytene piRNAs in mammals, has been hard to decipher because they appear to be rapidly diverging both between and within species. This study highlights the differential expression of piRNAs among mouse strains and reveals that novel endogenous retrovirus insertions can drive the production of novel piRNAs over extremely short evolutionary timescales. This discovery will be of great interest to biologists who study piRNAs, evolution, and mammalian spermatogenesis. Below, I list points that could be clarified in the text, one additional experimental analysis that would significantly enhance the work, and a few technical concerns that likely can be clarified without additional experimentation.

Major Comments

1. For pachytene piRNAs, e.g., pi-14-qA3-284, what is driving the differential expression? Are the piRNA-producing sequences different among the strains? Have these sequence differences accumulated faster in pachytene piRNA loci than in other non-coding RNAs? Are the sequences of the regulatory elements identical between strains? It is worth noting that both 14-qA3-284 and 14 qC1 1261 are mouse-specific piRNA genes (Ozata et al., RNA 2022). Pi-17-qE1.1-7037.1 (also mouse-specific) has been proposed to initiate piRNA production from 14-qA3-284, whereas 14-qC1-1261 initiates piRNA production from multiple, more deeply conserved piRNA genes (Ozata et al., Fig. 3D, 3E, and Table S2). Does a change in 14-qC1-1261 expression affect the piRNAs produced from those target genes?
2. What is the inter-animal variation in individual piRNA abundance within a strain? Is it always smaller than the variance between strains?
3. For piRNAs derived from mRNAs (e.g., Zbtb37), does the expression of the mRNA in long RNA-seq differ between strains? Should the piRNA data be normalized for any differences in mRNA expression?
4. Since the differential expression of piRNAs for Noct reflects the production of piRNAs from the first Noct intron in BL6 and NOD, does the IAP insertion lead to the conversion of Noct into a long-first-exon pachytene piRNA-producing locus (Yu et al., Nat Comm 2021)? I.e., is Noct spliced more efficiently in C3H and 129? This can be readily determined by measuring the ratio of intronic reads to exon1-exon2 junction reads in nuclear RNA (or

perhaps even in the authors existing long RNA-seq) or by using an analogous RT-PCR assay.

5. The authors write in the Discussion, that "What is unique to polymorphic piRNA producing loci is the magnitude of new material for natural selection to act upon. It is perhaps because of the many possibilities for positive or negative effects on fitness by each individual piRNA, among the many produced from a single locus, that piRNA-producing loci are gained and lost so fast during evolution." This is the most elegant description of what makes piRNAs so fascinating that I have ever read.

****Technical Comments****

1. Illumina's TruSeq small RNA library preparation kit is notorious for misrepresenting the relative abundance of individual sequences. This, of course, does not matter when identical sequences are compared between libraries, but it does matter when comparing the abundance of sequences from a specific piRNA-producing locus whose overall sequence has drifted between strains.

2. The authors report that for differential expression analysis, they used only "reads mapping to unique locations in the reference mouse genome." I don't understand how this strategy works when comparing multiple different strains whose genomes may not contain identical piRNA sequences. This is of particular importance because the authors are studying rapidly evolving loci.

Aren't the genomic sequences of each strain available? If not, then the sequences of the relevant loci should be reconstructed from the strain-specific long RNA-seq data and then then each strain's assembled transcripts used to map small RNAs from that strain.

3. Only piRNAs present, on average, at {greater than or equal to}10 molecules/cell can be compared in differential expression analysis, because piRNAs <10 molecule/cell are not likely to be produced in every cell. Consequently, differential expression may simply reflect stochastic sampling. Since the authors do not measure molecules of piRNA/cell, this can be addressed by resampling a single data set to determine the abundance threshold (in reads per million) that does not falsely produce differential expression.

****Minor Points****

1. What do the authors mean by "bidirectional clusters"? Does this refer to divergently transcribed piRNA-precursors-such as is found for many pachytene piRNA-producing loci- or to convergently transcribed ("dual-strand") clusters like those found in flies?

2. It would be helpful to include in the legends the number of independent samples that were analyzed for each strain (e.g., in Figure 1A) and whether the individual data points correspond to mean or median values.

3. For the differentially expressed loci highlighted in Figure 1, please (1) note whether the locus produces transposon-silencing piRNAs, pre-pachytene piRNAs from the 3' UTR of an mRNA, or pachytene piRNAs from one of the annotated pachytene piRNA genes; and (2) indicate which are evolutionarily conserved (i.e., present at the syntenic location) across placental mammals, only among rodents, and which are mouse-specific (e.g., 14-qA3-284 and 14-qC1-1261).

4. The authors write, "how different are the sets of piRNAs expressed in genetically different individuals of the same species? There are few studies addressing this question, none of which in mammals" and "it is unknown whether there are differences in the loci that produce piRNAs in different mice, or different individuals of any other mammalian species." Isn't this exactly what Ozata et al. (Nat Ecol & Evo 2020) showed for humans, that the sequences of pachytene piRNA loci were rapidly diverging among individual modern humans?

5. I am confused by the statement that "Although the piRNA pathway is conserved between flies and mice, the mechanisms of piRNA biogenesis are quite distinct." Only the mechanisms by which animals define piRNA-producing transcripts differ between species: e.g., flies mark piRNA-producing clusters with the heterochromatin-binding protein Rhino, whereas mammalian pachytene piRNA loci are euchromatic, have unusually long first exons, and are broadly bound by acylated histones and the protein BTBD18 (Yu et al., Nat Comm 2021). The rest of the mechanism for producing piRNA is conserved back to the last common ancestor of all animals.

6. Pachytene piRNAs are unlikely to play a role in transposon silencing (Wu et al., Nat Genet 2020; Choi et al., PLoS Genet 2021). Although transposon silencing in meiotic cells requires MIWI (Reuter et al., Nature 2011), the piRNAs that silence LINE1 elements in these cells do not derive from pachytene piRNA clusters.

****Referees cross-commenting****

1. The authors selected 19-36 nt small RNAs from their sequence libraries for their mapping analysis and considered those reads as piRNAs without examining whether they actually bind to PIWI proteins. I found this to be problematic because this fraction might contain a number of other small RNAs such as miRNAs as well as various RNA degradation products.

This seems highly unlikely given that the authors used the published annotations for the piRNA clusters, which do not overlap with any other genes or small RNA loci.

2. Figure 5A: The authors state that Noct mRNA expression is nearly identical among mouse strains, regardless of the insertion of IAP into the gene. However, Noct mRNA with

IAP sequence is constantly consumed as "piRNA" precursors upon transcription. Therefore, it is necessary to examine the mRNA levels after piRNA biogenesis in the testes is completely inhibited from the first step (right after transcription).

There is no evidence that piRNA production completely consumes piRNA precursor transcripts. Just the opposite: in mice, piRNA precursor transcripts are readily detected even for those that produce a high level of piRNAs. Moreover, mRNAs that produce piRNAs in flies still make enough mRNA to produce protein.

3. Page 5: "it looks like the IAP is causing the unspliced transcript to be exported from the nucleus and to be recognised as a piRNA precursor." I have heard of the phenomenon that inserting a TE sequence into a protein-coding mRNA can affect the splicing pattern of the host mRNA. However, the aberrant, unspliced mRNAs should be retained in the nucleus. It seems very unlikely to me that the insertion of IAP would result in active export of unspliced mRNA into the cytoplasm. This makes me doubt whether the authors' interpretation is correct. Also, does such splicing inhibition and promotion of pre-mRNA export by IAP insertion affect the expression level of NOCT protein? This needs to be examined.

Long first exons (i.e., intron retention) is associated with piRNA production in mice (e.g., Yu et al., Nat Comm 2021), Koalas (Yu et al., Cell 2019), and flies (Zhang et al., Cell 2014). The current model for piRNA production is that unspliced transcripts resemble retroviral genomic RNA and are therefore targeted for silencing by piRNAs. It is entirely consistent with the current state of knowledge in the piRNA field that unspliced RNA containing TE or ERV sequences will be exported to the cytoplasm.

4. I do not think the statement "IAP insertion in Noct is associated with post-transcriptional processing of germline- expressed transcripts into piRNAs" is accurate.

I think the authors data strongly support their assertion, and I do not understand the basis of this comment.

- Exclude data from "predicted piRNA clusters"

Why? This is one of the novel aspects of the manuscript and will help other labs in their study of piRNAs.

2. Significance:

Significance (Required)

The function of non-transposon-silencing piRNAs, particularly the pachytene piRNAs in mammals, has been hard to decipher because they appear to be rapidly diverging both between and within species. This study highlights the differential expression of piRNAs among mouse strains and reveals that novel endogenous retrovirus insertions can drive the production of novel piRNAs over extremely short evolutionary timescales. This discovery will be of great interest to biologists who study piRNAs, evolution, and mammalian spermatogenesis.

Phil Zamore (reviewer 1)

3. How much time do you estimate the authors will need to complete the suggested revisions:

Estimated time to Complete Revisions (Required)

(Decision Recommendation)

Between 1 and 3 months

4. Review Commons values the work of reviewers and encourages them to get credit for their work. Select 'Yes' below to register your reviewing activity at Web of Science Reviewer Recognition Service (formerly Publons); note that the content of your review will not be visible on Web of Science.

No

Review #2

1. Evidence, reproducibility and clarity:

Evidence, reproducibility and clarity (Required)

In this manuscript, the authors deep-sequenced small RNAs from the germline of genetically distinct male mice, from which they selected 19-36 nt read sequences, bioinformatically analyzed, and compared them across strains. As a result, the authors argued that there were significant differences in piRNA production depending on the presence and absence of endogenous retroviral insertions at the loci that produce piRNAs. The authors also argued that the findings provide evidence that transposable elements may contribute to the rapid evolution of piRNA-producing loci in mammals.

The problems the authors tried to tackle in this study are important in the field of piRNA. I thank the authors for their research efforts, but at the same time I find several issues to be addressed before this paper can be considered for publication.

1. The results shown in Figures 1D and 3A indicate that only a small fraction of 19-36 nt small RNAs were mapped to the Noct gene in C3H and 129 (for example), but the mapping rate increased significantly in BL6 and NOD. The obvious difference the authors found between the two groups was whether the Noct locus had an IAP insertion (for BL6 and NOD) or not (for C3H and 129). This suggests that the IAP insertion has ability to convert a protein-coding gene into a locus that produces 19-36 nt small-RNAs. The authors then extended their analysis to other loci in other genetically distinct mice, including known and predicted piRNA-producing loci, and argued that the above conclusion can be applied to those loci. However, the details (e.g. genome mapping) were not available in this manuscript. Thus, I cannot determine whether the authors' claims are correct. Please show mapping data of other loci in other genetically distinct mice, including known piRNA-producing loci.

2. Figures 2D-2G show small RNA mapping data. However, I found very difficult to interpret or to evaluate these results because the necessary information is not provided, for example, whether the predicted piRNA cluster PTc927 has ERV (or any other TE) insertions only in C3H (Figure 2D).

3. The authors selected 19-36 nt small RNAs from their sequence libraries for their mapping analysis and considered those reads as piRNAs without examining whether they actually bind to PIWI proteins. I found this to be problematic because this fraction might contain a number of other small RNAs such as miRNAs as well as various RNA degradation products. PIWI immunoprecipitation may be the solution.

4. The way small RNA mapping data is shown in Figures 1D and 3A is different. What is the reason for this? If there is no special reason, please show data in the same way.

5. Figure 3B: The authors found a good correlation between "black" and "white" genotypes (according to the legend in Fig 3B) and the number of 19-36 nt small RNAs mapped to the Noct locus. However, it remains to be examined whether such a correlation can be applied to "gray" genotype. This needs to be done.

6. Figure 4A: The authors examined "clusters with indels" and "clusters without indels". However, the data should be heavily influenced by which strain of mice the author used as the reference for their analysis. For example, the Noct locus had an IAP insertion in BL6 but not in C3H. If the authors consider BL6 as a reference, C3H has an IAP insertion. If the authors consider C3H as a reference, BL6 has an IAP deletion. How are these handled?

7. Figure 5A: The authors state that Noct mRNA expression is nearly identical among

mouse strains, regardless of the insertion of IAP into the gene. However, Noct mRNA with IAP sequence is constantly consumed as "piRNA" precursors upon transcription. Therefore, it is necessary to examine the mRNA levels after piRNA biogenesis in the testes is completely inhibited from the first step (right after transcription).

8. Page 5: "it looks like the IAP is causing the unspliced transcript to be exported from the nucleus and to be recognised as a piRNA precursor." I have heard of the phenomenon that inserting a TE sequence into a protein-coding mRNA can affect the splicing pattern of the host mRNA. However, the aberrant, unspliced mRNAs should be retained in the nucleus. It seems very unlikely to me that the insertion of IAP would result in "active" export of unspliced mRNA into the cytoplasm. This makes me doubt whether the authors' interpretation is correct. Also, does such splicing inhibition and promotion of pre-mRNA export by IAP insertion affect the expression level of NOCT protein? This needs to be examined.

I suggest that the authors follow additional suggestions when revising the manuscript. Some of these may overlap with the suggestions made above.

- Exclude data from "predicted piRNA clusters"
- Concentrate on four mice strains, BL6, NOD, C3H, and 129
- Show detailed mapping data for the five piRNA-producing loci in Figure 1C
- Show TE insertions in the five loci
- ChIP data is not required

Do you have suggestions that would help the authors improve the presentation of their data and conclusions?

1. In Figure 1, known piRNA clusters are analyzed, and in Figure 2, new piRNA clusters are analyzed. I would like to see more information on the relationship between them. For example, how many of the de novo piRNA clusters identified in Figure 2 were known in Figure 1?
2. Figure 2B: Please state that this is about a total of six pairwise comparisons made between four different testis strains.
3. Figures 2D and 2E: Please indicate what percentage of each type of piRNA cluster is present.
4. Figure 3A: Please describe exon, intron, UTR, splicing variant, and repeat (Figure 3A) in the text. What do brown piRNAs out of piRNA mapping indicate?

2. Significance:

Significance (Required)

- Describe the nature and significance of the advance (e.g. conceptual, technical, clinical) for the field.

The problems they tried to tackle in this study are important in the field of piRNA.

- Place the work in the context of the existing literature (provide references, where appropriate).

It is well known in the field that the identity of piRNA clusters is acquired by Rhino, H3K9me3, etc. from fly analysis, and furthermore, that new insertion of piRNA is related to piRNA production. On the other hand, it is a unique approach and very interesting to compare this issue in mammal and by mouse subspecies.

- State what audience might be interested in and influenced by the reported findings.

piRNA researchers. Embryologists with a focus on the testis.

- Define your field of expertise with a few keywords to help the authors contextualize your point of view. Indicate if there are any parts of the paper that you do not have sufficient expertise to evaluate.

Reproductive epigenome

3. How much time do you estimate the authors will need to complete the suggested revisions:

Estimated time to Complete Revisions (Required)

(Decision Recommendation)

Between 3 and 6 months

4. Review Commons values the work of reviewers and encourages them to get credit for their work. Select 'Yes' below to register your reviewing activity at Web of Science Reviewer Recognition Service (formerly Publons); note that the content of your review will not be visible on Web of Science.

Yes

Full Revision

Manuscript number: RC-2022-01746

Corresponding author(s): Tanya, Vavouri; Sonia, Forcales; Josep C., Jimenez-Chillaron

[Please use this template only if the submitted manuscript should be considered by the affiliate journal as a full revision in response to the points raised by the reviewers.]

*If you wish to submit a preliminary revision with a revision plan, please use our "Revision Plan" template. **It is important to use the appropriate template to clearly inform the editors of your intentions.**]*

1. General Statements

We appreciate the feedback provided by the two reviewers. Their insightful comments have helped us improve the manuscript. Here, we address each comment and detail the corresponding modifications made in the revised manuscript.

In addition, we would like to note the following change to the author list. All the analyses, figures and tables included in the revised manuscript were done by Adrià Mitjavila and therefore he is now included in the author list as co-first author, together with Eduard Casas and Pío Sierra.

Reviewer #1 (Evidence, reproducibility and clarity (Required)):

This manuscript was a pleasure to read. The function of non-transposon-silencing piRNAs, particularly the pachytene piRNAs in mammals, has been hard to decipher because they appear to be rapidly diverging both between and within species. This study highlights the differential expression of piRNAs among mouse strains and reveals that novel endogenous retrovirus insertions can drive the production of novel piRNAs over extremely short evolutionary timescales. This discovery will be of great interest to biologists who study piRNAs, evolution, and mammalian spermatogenesis. Below, I list points that could be clarified in the text, one additional experimental analysis that would significantly enhance the work, and a few technical concerns that likely can be clarified without additional experimentation.

Major Comments

(1) For pachytene piRNAs, e.g., pi-14-qA3-284, what is driving the differential expression? Are the piRNA-producing sequences different among the strains?

*To address technical comment (2), about the issue of mapping the small RNA data of all strains to the reference mouse genome, we modified our original methodology. We now map the small RNA data of each strain to the corresponding strain genome (see **Methods**). This way we avoid possible artefacts caused by reads mapping to positions with polymorphisms between strains, that may not have previously mapped or falsely mapped at a different locus due to sequence differences with the reference genome. However, an issue with using the strain genomes is that the strain genomes contain more gaps than the reference mouse genome. As a consequence, some of the clusters previously found to be differentially expressed are not in the new list due to gaps in strain genomes and/or other problems in aligning parts of the different mouse strains. piRNA cluster pi-14-qA3-284 mentioned in this comment is one of them.*

*Here, we address the general question of whether, for pachytene piRNAs, differences in sequences of piRNA-producing genes drive differential expression. To do this, we compared sequence similarity (% identity) between piRNA producing genes that are differentially expressed between strains against that of piRNA producing genes that are not differentially expressed between strains. We found that on average differentially expressed piRNA producing loci have more sequence changes than the rest, but, for most comparisons, the differences are not statistically significant (wilcoxon rank sum test, significance threshold 0.05, new **Supplementary Fig 6A**). Note that after selecting only pachytene piRNA clusters and splitting them into differentially and not-differentially expressed, the number of observations for some of the groups is only between 4 and 7. The definition of pachytene piRNA clusters is from Li et al (Li et al, 2013) and corresponds to strain BL6, therefore for this analysis we used only pairwise comparisons against BL6. Also, since pachytene piRNAs are enriched in whole testes samples and not in spermatogonia, we did this analysis only for strains from which we have small RNA data from testes (i.e. not for strain CAST). Taking all this into account, we conclude that differentially expressed pachytene piRNA producing loci tend to have more sequence differences than other piRNA producing loci and this trend is compatible with sequence differences driving differential expression, with the limitation that the number of observations is low. We currently do not know whether the relationship between the two variables (sequence change and expression change) is causal.*

Have these sequence differences accumulated faster in pachytene piRNA loci than in other non-coding RNAs?

To test whether more sequence differences have accumulated in pachytene piRNA loci than in other non-coding RNAs, we compared sequence similarity (% identity) between known piRNA-producing loci from Li et al (2013), predicted piRNA-producing loci and a random set of long non-coding RNAs. Protein-coding genes were removed from all three classes since genetic variation is less common in coding exons and can bias the analysis. Furthermore, the vast majority of protein-coding transcripts producing piRNAs are classed as prepachytene piRNA clusters (Li et al, 2013). Therefore, by removing protein-coding genes from known and predicted piRNA-producing loci, we have narrowed down the set to almost exclusively pachytene piRNAs. We found that piRNA-producing loci have lower % identity (i.e. they have accumulated more

sequence differences) compared to other non-coding RNAs, suggesting that, during evolution, sequence differences accumulate faster in pachytene piRNA loci (see new **Supplementary Fig 5** of the revised manuscript).

In the revised manuscript we clarify this in the 3rd paragraph of the results section:

“Between mouse strains, piRNA-producing loci (excluding those also coding for protein-coding genes) have more sequence differences compared to other long non-coding RNAs (**Supplementary Fig 5**).”

Are the sequences of the regulatory elements identical between strains?

To address this comment, first, we compared the similarity of the promoter regions (5,000bp centred at the start) of differentially expressed piRNA-producing loci against those of not differentially expressed loci. In all pairwise comparisons, the promoters of differentially expressed pachytene piRNA-producing loci have more sequence differences than the promoters of other pachytene piRNA-producing loci – although not all p-values reach significance (**Supplementary Fig 6B**). Second, we tested whether piRNA clusters with different expression between strains contain mutations that affect the binding of transcription factors in their promoters. To address this, we retrieved ChIP-seq A-MYB and TCFL5 data for strain BL6 from (Yu et al, 2023) and identified those that are in promoter regions (+/- 2,500bp from TSS) of pachytene piRNA clusters from (Li et al, 2013). We then split these into clusters with / without significant differential expression in testis between BL6 and other strains and compared the % identity between the two groups. Although the number of observations was too low to reach general conclusions, again we found that the A-MYB and TCFL5 bound regions of promoters of differentially expressed piRNA clusters often contain more sequence differences than those of promoters of similarly expressed piRNA clusters (**Supplementary Fig 6C**). We conclude that the regulatory regions of piRNA-producing loci are not always identical and that those with more sequence differences are often those differentially expressed between strains.

In the revised manuscript we clarify this in the 3rd paragraph of the results section:

“Focusing on pachytene piRNA producing loci that are the ones most abundant in whole testes, and comparing BL6 to the other three strains, we found that differentially expressed loci tend to have more sequence differences along the entire locus (**Supplementary Fig 6A**), at the promoter (**Supplementary Fig 6B**) and at transcription factor binding sites in promoters (**Supplementary Fig 6C**), although there is a low number of observations and not all comparisons reach statistical significance.”

It is worth noting that both 14-qA3-284 and 14 qC1 1261 are mouse-specific piRNA genes (Ozata et al., RNA 2022). Pi-17-qE1.1-7037.1 (also mouse-specific) has been proposed to initiate piRNA production from 14-qA3-284, whereas 14-qC1-1261 initiates piRNA production from multiple, more deeply conserved piRNA genes (Ozata et al., Fig. 3D, 3E, and Table S2).

Does a change in 14-qC1-1261 expression affect the piRNAs produced from those target genes?

*In the revised version of the manuscript, 14-qC1-1261 is no longer among the piRNA clusters analysed because in NOD, the strain in which it appeared more expressed in the original version of the paper, there is a 3.6Mb gap (see grey bar in the figure below) in the assembly at the position where it is expected to be (highlighted in blue in BL6 and NOD in **Fig R1**, below). Nevertheless, we can address this comment using the results from analysing the data after mapping reads to BL6, from the first version of the manuscript. According to Yu et al (Yu et al, 2023), piRNA cluster 14-qC1-1261 triggers piRNA production from several piRNA clusters, namely 1-qC1.3-637, 3-qA2-617, 4-qC5-17839, 5-qF-14224, 5-qG2-950, 9-qC-10667, 10-qB4-6488, 14-qA3-19970, 14-qC1-1010, 15-qD1-17920, 15-qD1-4001 and 18-qE1-36451. None of these targeted piRNA clusters was found to be differentially expressed in any of the pairwise strain comparisons (results of differential expression between strains shown in the original version of **Supplementary Table S4**). Therefore, we have no evidence that an increase in piRNA expression from one cluster translates into differential expression of these predicted targeted piRNA clusters.*

Figure R1. A gap in NOD strain assembly at the genomic position expected to find piRNA cluster 14-qC1-1261. The position where we would expect to find the cluster is highlighted in light blue, gaps are shown in grey and piRNA clusters in black.

(2) What is the inter-animal variation in individual piRNA abundance within a strain? Is it always smaller than the variance between strains?

Yes, inter-animal variation is smaller than inter-strain variation. This is shown in the heatmaps in **Fig 1B** and **Fig 2A**, where samples (columns) of the same strain always cluster together and separately from samples of other strains. It can also be seen in **Fig 1C**, where each point represents a sample and again all samples from the same strain cluster together and separate from samples from the other strains. In addition, we have added a figure with all the pair-wise scatterplots and correlations of expression of piRNA-producing loci between all samples in the supplementary material (new **Supplementary Fig 3**).

We comment on this in the second paragraph of the results:

“Despite the high concordance of piRNA expression between all the studied mice, the abundance of piRNAs was more similar between animals of the same inbred strain than animals of different strains (Fig 1B, Supplementary Fig 3).”

(3) For piRNAs derived from mRNAs (e.g., Zbtb37), does the expression of the mRNA in long RNA-seq differ between strains? Should the piRNA data be normalized for any differences in mRNA expression?

We sequenced and analysed total RNA data from the same testis RNA samples from which we sequenced small RNAs (**Supplementary Tables S13-S17 and Fig 5B-C**). Neither Zbtb37 nor Noct show significant differences in expression based on RNA-seq data but both show differences in small RNA production (see **Fig R2, below and Fig 5C**). Of the three protein-coding piRNA-producing genes shown in **Fig 1C** (Noct, Zbtb37 and Mrs2), only Mrs2 shows coordinated increase in mRNA expression and piRNA abundance in the four mouse strains. Given the lack of general correlation, we decided to keep the measurements of small RNA data and long RNA data separate.

Figure R2. Comparison of fold change in piRNA-precursor expression (total RNA-seq) versus fold change in small RNA abundance in pairwise comparisons of the four inbred mouse strains. Points represent the known piRNA-producing loci from Li et al. Three protein-coding genes producing piRNAs (those highlighted in Fig 1C) are marked in red.

In the revised manuscript, we have added two new panels to **Fig 5**, showing that the Noct piRNA precursor is not differentially expressed between the four strains and the following sentence in the penultimate paragraph of the results:

“We also sequenced and analysed Noct expression in testes of the four inbred strains. None of the comparisons revealed a significant difference in Noct expression (Fig 5B-C and Supplementary Tables S13-S17)”

(4) Since the differential expression of piRNAs for Noct reflects the production of piRNAs from the first Noct intron in BL6 and NOD, does the IAP insertion lead to the conversion of Noct into a long-first-exon pachytene piRNA-producing locus (Yu et al., Nat Comm 2021)? I.e., is Noct spliced more efficiently in C3H and 129? This can be readily determined by measuring the ratio of intronic reads to exon1-exon2 junction reads in nuclear RNA (or perhaps even in the authors' existing long RNA-seq) or by using an analogous RT-PCR assay.

*First, we want to clarify that the piRNA cluster overlapping Noct, in strain BL6, which has the IAP, has peak expression at pre-pachytene stages of sperm development and is classed as a pre-pachytene cluster, even though it retains some expression at later stages (see **Fig 5A**).*

*Regarding the question about the differences in splicing of the two Noct alleles, we used total (long) RNA-seq data to evaluate the relative presence of the Noct intron in sequenced long transcripts. We analysed three datasets; (i) testis RNA-seq data from fourteen genotyped ICR mice (9 Noct IAP+ and 5 Noct IAP-) (ii) testis RNA-seq data from strains BL6 (Noct IAP+ strain), NOD (Noct IAP+ strain), 129 (Noct IAP- strain), C3H (Noct IAP- strain) and (iii) testis RNA-seq data from Yu et al, Cell, 2019 from C57BL/6J (Noct IAP+ strain), C57BL/6NJ (Noct IAP+ strain), AKR/J (Noct IAP+ strain), C3H/HeJ (Noct IAP- strain) and LP/J (Noct IAP- strain). We considered two metrics of splicing; first, splicing efficiency measured as the ratio of split versus unsplit reads at splice junctions and second relative intron expression measured as the ratio of read coverage of the intron to that of the surrounding exons. Reads mapping to repeats and sequencing gaps were excluded to avoid mapping artefacts. Using both metrics, the dataset of 14 ICR genotyped mice shows significantly higher intron retention in mice with the IAP compared to mice without it (new **Supplementary Fig 9, top panels**). For the other datasets, the trend is weaker (new **Supplementary Fig 9, middle and lower panels**).*

(5) The authors write in the Discussion, that "What is unique to polymorphic piRNA producing loci is the magnitude of new material for natural selection to act upon. It is perhaps because of the many possibilities for positive or negative effects on fitness by each individual piRNA, among the many produced from a single locus, that piRNA-producing loci are gained and lost so fast during evolution." This is the most elegant description of what makes piRNAs so fascinating that I have ever read.

We thank the reviewer for the positive comment on the discussion.

Technical Comments

(1) Illumina's TruSeq small RNA library preparation kit is notorious for misrepresenting the relative abundance of individual sequences. This, of course, does not matter when identical sequences are compared between libraries, but it does matter when comparing the abundance of sequences from a specific piRNA-producing locus whose overall sequence has drifted between strains.

Although we cannot discount the possibility that some differences in expression may be due to differences in sequence causing misrepresentation of the abundance of individual sequences, we argue that the findings of our study are unlikely to be severely affected but sequence biases primarily because we have analysed the expression of piRNA clusters, not individual piRNAs, so their expression values should be less affected by those of individual piRNAs with sequence differences between strains. Also, as explained in the methods, reads mapping to repeats, polymorphic or fixed, were not used for the analysis of differential piRNA expression.

We have included the following sentence on this matter, in the methods section on RNA-sequencing:

"It should be noted that sequence differences between the strains may also affect the adapter ligation and PCR amplification steps of the small RNA sequencing protocols, leading to distortion in abundance of individual piRNAs between strains. However, this limitation is unlikely to affect differential expression of piRNA clusters because the latter usually consist of hundreds of different piRNAs, most of which have the same sequence between strains. Also, reads mapping to repeats, polymorphic or fixed, were not used for the analysis of differential piRNA expression."

(2) The authors report that for differential expression analysis, they used only "reads mapping to unique locations in the reference mouse genome." I don't understand how this strategy works when comparing multiple different strains whose genomes may not contain identical piRNA sequences. This is of particular importance because the authors are studying rapidly evolving loci.

Aren't the genomic sequences of each strain available? If not, then the sequences of the relevant loci should be reconstructed from the strain-specific long RNA-seq data and then each strain's assembled transcripts used to map small RNAs from that strain.

In the original version of the manuscript, we had mapped all small RNA data from all strains to the reference mouse genome. Since the genomes of the inbred mouse strains used in this paper (or of very related strains) are publicly available, we followed the reviewer's suggestion and reanalysed the data using the strain genomes. We mapped all the small RNA data to the nearest publicly available strain genome and reanalysed the resulting piRNA cluster expression. Although the expression counts and, some of the differentially expressed clusters have changed, the main conclusions remain the same. The new small RNA strategy is explained in the Methods and all figures, results and tables have been updated.

(3) Only piRNAs present, on average, at {greater than or equal to}10 molecules/cell can be compared in differential expression analysis, because piRNAs <10 molecule/cell are not likely to be produced in every cell. Consequently, differential expression may simply reflect stochastic sampling. Since the authors do not measure molecules of piRNA/cell, this can be addressed by resampling a single data set to determine the abundance threshold (in reads per million) that does not falsely produce differential expression.

We followed the reviewer's advice to test whether stochastic sampling of lowly expressed molecules may cause false positive differential expression detection. We carried out a randomisation, as suggested by the referee. We shuffled the labels of the samples so that we randomly picked 4 testis small RNA samples, out of the 10 samples from inbred mouse strains, and tested for differential expression of 2 versus 2 samples. All the rest of the steps of the analysis were identical to the ones in the manuscript. We repeated the shuffling of sample labels and the test of differential expression 100 times. We then counted how often each piRNA cluster was detected as differentially expressed in 100 randomisations. As shown in **Figure R3** below, we did not find that that lowly expressed clusters are significantly differentially expressed by chance more often than other, more highly expressed clusters. Manually inspecting some of those that are often differentially expressed between the randomly picked samples, we saw that they are clusters that have extremely low expression in all samples of one strain in comparison to the other three strains. One such example is 10-qC1-2617 which has normalized count of 0 in the three samples of strain 129 and high counts in all the other samples from the other three strains (see **Fig 1C** and **Supplementary Table S3**).

The reasons why lowly expressed clusters are not often detected as differentially expressed are likely the following: First, lowly expressed clusters are composed of multiple piRNAs, so the effect of stochastic sampling of individual piRNA molecules is buffered. Second, DESeq2, the tool that we used to identify differentially expressed clusters, performs independent filtering based on the mean of normalized counts so that lowly expressed clusters with high dispersion are not tested for differential expression.

Figure R3. Analysis of the relationship between piRNA cluster expression and likelihood of being predicted to be differentially expressed in 100 sample randomisations.

Minor Points

(1) What do the authors mean by "bidirectional clusters"? Does this refer to divergently transcribed piRNA-precursors such as is found for many pachytene piRNA-producing loci or to convergently transcribed ("dual-strand") clusters like those found in flies?

We used the term as defined by default by proTRAC, the tool that we used to predict piRNA clusters. When at least 10% of the reads map antisense to the "main strand", i.e. the strand where most reads map, then proTRAC annotates this cluster as bidirectional. ProTRAC does not take into account whether the precursor transcripts are divergently or convergently transcribed.

We have clarified the definition of bidirectional cluster in the Method section:

"Clusters were annotated as mono-directional or bi-directional by proTRAC, depending on whether at least 10% of the reads mapping to the locus were on the antisense strand."

(2) It would be helpful to include in the legends the number of independent samples that were analyzed for each strain (e.g., in Figure 1A) and whether the individual data points correspond to mean or median values.

We have clarified this in the legend of **Fig 1**:

"Each point represents the mean of the log2 transformed, scaled read counts of all samples of the same strain (3 samples each, for strains C3H and 129 and 2 samples each, for strains NOD and BL6)."

(3) For the differentially expressed loci highlighted in Figure 1, please (1) note whether the locus produces transposon-silencing piRNAs, pre-pachytene piRNAs from the 3' UTR of an mRNA, or pachytene piRNAs from one of the annotated pachytene piRNA genes; and (2) indicate which are evolutionarily conserved (i.e., present at the syntenic location) across placental mammals, only among rodents, and which are mouse-specific (e.g., 14-qA3-284 and 14-qC1-1261).

We have used the percentage of uniquely mapping reads as a proxy for whether the locus produces transposon-silencing piRNAs, since transposon-silencing piRNAs map to multiple sites. We have included this information as supplementary tables. **Supplementary Table S11** contains the percentage of reads that map uniquely to the known Li et al piRNA clusters. **Supplementary Table S12** contains the percentage of reads that map uniquely to the predicted piRNA clusters.

We have added **Supplementary Fig 2**, where we show the conservation of all 214 piRNA clusters analysed in **Fig 1** in rat, other placental mammals (macaque, human, marmoset and opossum) and in the egg-laying mammal platypus. Evolutionary conservation and piRNA expression in these species were retrieved from Ozata et al (Özata et al, 2020). Also, in this

figure, we split the clusters into prepachytene, hybrid and pachytene, using the annotation from Li et al (Li et al, 2013). With few exceptions (like pi-Noct), most piRNAs produced from protein-coding genes are produced from their 3' UTR and the vast majority of piRNA precursors that are protein-coding genes are annotated as prepachytene (Li et al, 2013). Therefore, whether a locus produces piRNAs from the 3' UTR of an mRNA or not, is mostly explained by the prepachytene-hybrid-pachytene classification of piRNAs.

In addition, in the 2nd paragraph of the results, we note the conservation of the clusters shown in **Fig 1C:**

"The protein-coding gene Nocturnin (Noct) is a mouse-specific prepachytene piRNA cluster (Li et al, 2013) that produces abundant piRNAs in BL6 and NOD strains but not in C3H and 129 (Fig 1C,D, Supplementary Fig 2 and Supplementary Fig 4A). The gene Zbtb37, a prepachytene piRNA cluster that also produces piRNAs in human, macaque and platypus, produces significantly more piRNAs in strains NOD, C3H and 129 than in BL6 (Fig 1C, Supplementary Fig 2 and Supplementary Fig 4B). The gene Mrs2, which is a hybrid, mouse-specific piRNA cluster, produces piRNAs in three mouse strains but nearly none in NOD (Fig 1C, Supplementary Fig 2 and Supplementary Fig 4C). Last, an intergenic, pachytene, cluster on chromosome 10 that is not conserved in other species, produces nearly few piRNAs in strain 129, many piRNAs in strains NOD and C3H and still twice as many in BL6 (Fig 1C, Supplementary Fig 2 and Supplementary Fig 4D)."

(3) The authors write, "how different are the sets of piRNAs expressed in genetically different individuals of the same species? There are few studies addressing this question, none of which in mammals" and "it is unknown whether there are differences in the loci that produce piRNAs in different mice, or different individuals of any other mammalian species." Isn't this exactly what Ozata et al. (Nat Ecol & Evo 2020) showed for humans, that the sequences of pachytene piRNA loci were rapidly diverging among individual modern humans?

We thank the reviewer for pointing out that this part of the introduction was not clear. We have edited. Indeed, Ozata et al showed that piRNA producing loci are genetically different between different individuals of the same species, humans and that the expression of piRNAs from different individuals could be clustered into different groups. Having said that, a link between genetic and expression different was not made in Ozata et al. piRNAs from several human samples were analysed, and the samples were clustered into three groups based on piRNA expression, but the authors stated that possible causes for these differences in piRNA expression were defects in spermatogenesis (azoospermia) and/or mutagens or chemotherapy rather than natural genetic variation between healthy individuals.

We have clarified the findings of the study by Ozata in the 2nd and 3rd paragraphs of the introduction:

"Within the human population, piRNA-producing loci contain more genetic variants than any other transcribed genomic feature (Özata et al, 2020). These independent pieces of evidence suggest that there is little purifying selection pressure on the DNA sequence of piRNA-

producing loci. Given that approximately half of all mammalian mouse piRNA precursors are transcripts of protein-coding genes, it is remarkable how evolvable piRNA-producing loci are.

Considering the fast evolution of piRNAs and piRNA-producing genes, how different is the expression of piRNAs in genetically different, healthy individuals of the same species? There are few studies addressing this question. Differences in expression of piRNAs have been found between human individuals, however it is unclear whether this is due to differences in health, exposures to genotoxic agents or genetics (Özata et al, 2020). Under controlled environmental conditions, analyses of piRNAs from different *Drosophila* (Kelleher & Barbash, 2013; Shpiz et al, 2014) and zebrafish strains (Kaaij et al, 2013) have revealed that the identity of piRNA-producing loci and their expression levels vary depending on their genetic background.”

(4) I am confused by the statement that "Although the piRNA pathway is conserved between flies and mice, the mechanisms of piRNA biogenesis are quite distinct." Only the mechanisms by which animals define piRNA-producing transcripts differ between species: e.g., flies mark piRNA-producing clusters with the heterochromatin-binding protein Rhino, whereas mammalian pachytene piRNA loci are euchromatic, have unusually long first exons, and are broadly bound by acylated histones and the protein BTBD18 (Yu et al., Nat Comm 2021). The rest of the mechanism for producing piRNA is conserved back to the last common ancestor of all animals.

We have added that differences include the marks of piRNA-producing loci and their transcription.

“Although the piRNA pathway is conserved between flies and mice, some aspects of piRNA biogenesis are distinct, such as how piRNA-producing loci are marked in the genome and how they are transcribed (e.g. (ElMaghraby et al, 2019; Kneuss et al, 2019; Yu et al, 2021)”

(5) Pachytene piRNAs are unlikely to play a role in transposon silencing (Wu et al., Nat Genet 2020; Choi et al., PLoS Genet 2021). Although transposon silencing in meiotic cells requires MIWI (Reuter et al., Nature 2011), the piRNAs that silence LINE1 elements in these cells do not derive from pachytene piRNA clusters.

****Referees cross-commenting****

1) The authors selected 19-36 nt small RNAs from their sequence libraries for their mapping analysis and considered those reads as piRNAs without examining whether they actually bind to PIWI proteins. I found this to be problematic because this fraction might contain a number of other small RNAs such as miRNAs as well as various RNA degradation products.

This seems highly unlikely given that the authors used the published annotations for the piRNA clusters, which do not overlap with any other genes or small RNA loci.

2) Figure 5A: The authors state that *Noct* mRNA expression is nearly identical among mouse strains, regardless of the insertion of IAP into the gene. However, *Noct* mRNA with IAP sequence is constantly consumed as "piRNA" precursors upon transcription. Therefore, it is necessary to examine the mRNA levels after piRNA biogenesis in the testes is completely inhibited from the first step (right after transcription).

There is no evidence that piRNA production completely consumes piRNA precursor transcripts. Just the opposite: in mice, piRNA precursor transcripts are readily detected even for those that produce a high level of piRNAs. Moreover, mRNAs that produce piRNAs in flies still make enough mRNA to produce protein.

3) Page 5: "it looks like the IAP is causing the unspliced transcript to be exported from the nucleus and to be recognised as a piRNA precursor." I have heard of the phenomenon that inserting a TE sequence into a protein-coding mRNA can affect the splicing pattern of the host mRNA. However, the aberrant, unspliced mRNAs should be retained in the nucleus. It seems very unlikely to me that the insertion of IAP would result in active export of unspliced mRNA into the cytoplasm. This makes me doubt whether the authors' interpretation is correct. Also, does such splicing inhibition and promotion of pre-mRNA export by IAP insertion affect the expression level of NOCT protein? This needs to be examined.

Long first exons (i.e., intron retention) is associated with piRNA production in mice (e.g., Yu et al., Nat Comm 2021), Koalas (Yu et al., Cell 2019), and flies (Zhang et al., Cell 2014). The current model for piRNA production is that unspliced transcripts resemble retroviral genomic RNA and are therefore targeted for silencing by piRNAs. It is entirely consistent with the current state of knowledge in the piRNA field that unspliced RNA containing TE or ERV sequences will be exported to the cytoplasm.

4) I do not think the statement "IAP insertion in *Noct* is associated with post-transcriptional processing of germline- expressed transcripts into piRNAs" is accurate.

I think the authors data strongly support their assertion, and I do not understand the basis of this comment.

- Exclude data from "predicted piRNA clusters"

Why? This is one of the novel aspects of the manuscript and will help other labs in their study of piRNAs.

Reviewer #1 (Significance (Required)):

The function of non-transposon-silencing piRNAs, particularly the pachytene piRNAs in mammals, has been hard to decipher because they appear to be rapidly diverging both between and within species. This study highlights the differential expression of piRNAs among mouse strains and reveals that novel endogenous retrovirus insertions can drive the production of novel piRNAs over extremely short evolutionary timescales. This discovery will be of great interest to biologists who study piRNAs, evolution, and mammalian spermatogenesis.

Phil Zamore (reviewer 1)

Reviewer #2 (Evidence, reproducibility and clarity (Required)):

In this manuscript, the authors deep-sequenced small RNAs from the germline of genetically distinct male mice, from which they selected 19-36 nt read sequences, bioinformatically analyzed, and compared them across strains. As a result, the authors argued that there were significant differences in piRNA production depending on the presence and absence of endogenous retroviral insertions at the loci that produce piRNAs. The authors also argued that the findings provide evidence that transposable elements may contribute to the rapid evolution of piRNA-producing loci in mammals.

The problems the authors tried to tackle in this study are important in the field of piRNA. I thank the authors for their research efforts, but at the same time I find several issues to be addressed before this paper can be considered for publication.

1) The results shown in Figures 1D and 3A indicate that only a small fraction of 19-36 nt small RNAs were mapped to the Noct gene in C3H and 129 (for example), but the mapping rate increased significantly in BL6 and NOD. The obvious difference the authors found between the two groups was whether the Noct locus had an IAP insertion (for BL6 and NOD) or not (for C3H and 129). This suggests that the IAP insertion has ability to convert a protein-coding gene into a locus that produces 19-36 nt small-RNAs. The authors then extended their analysis to other loci in other genetically distinct mice, including known and predicted piRNA-producing loci, and argued that the above conclusion can be applied to those loci. However, the details (e.g. genome mapping) were not available in this manuscript. Thus, I cannot determine whether the authors' claims are correct. Please show mapping data of other loci in other genetically distinct mice, including known piRNA-producing loci.

*For each known piRNA-producing locus highlighted in **Fig 1C**, we provide small RNA mapping data along the locus, for each replicate of each strain, in **Supplementary Fig 4**.*

2) Figures 2D-2G show small RNA mapping data. However, I found very difficult to interpret or to evaluate these results because the necessary information is not provided, for example, whether the predicted piRNA cluster PTc927 has ERV (or any other TE) insertions only in C3H

(Figure 2D).

In **Supplementary Table 8**, we provide for each predicted piRNA cluster a binary classification (True / False) of whether it contains a LINE, SINE, ERV or IAP (type of ERV) that is inserted or deleted between any pair of the strains studied in this manuscript.

3) The authors selected 19-36 nt small RNAs from their sequence libraries for their mapping analysis and considered those reads as piRNAs without examining whether they actually bind to PIWI proteins. I found this to be problematic because this fraction might contain a number of other small RNAs such as miRNAs as well as various RNA degradation products. PIWI immunoprecipitation may be the solution.

We understand the concern of the reviewer and accept that the small RNA libraries likely also contain other types of noncoding RNAs. However, this should not affect the analyses and results described in the manuscript because it is all based on the analysis of small RNAs mapping to either known piRNA-producing loci from Li et al (Li et al, 2013) or to loci predicted by proTRAC to produce piRNAs. proTRAC defines these loci based on the length and nucleotide composition of the small RNAs that map to them, and these small RNAs have to have characteristics of piRNAs: according to the default proTRAC options that we used, 75% of small RNAs mapping to the locus need to be 24-32nt long and have T at nucleotide 1 or A at nucleotide 10. MicroRNAs are shorter and degradation products are without the 1T/10A bias. In addition, to address the concern of the referee, we tested the overlap between the predicted piRNA-producing loci and known noncoding RNAs. Of 851 piRNA-producing loci in alignable parts of the genomes of the five strains, only 37 overlap with annotated noncoding genes in C57BL6 and all 37 are lncRNAs and lincRNAs, not miRNAs.

4) The way small RNA mapping data is shown in Figures 1D and 3A is different. What is the reason for this? If there is no special reason, please show data in the same way.

In the previous version of the manuscript, in **Fig 3A** we had excluded multi-mapping reads. Now, for consistency, we include multi-mapping reads in both figures (multimapping reads are still not used for differential expression analysis). Regarding the difference in how the small RNA data is represented (lines versus rectangles) we kept this in the new version because we believe it is justifiable given the data and available space in the figures and we don't think that it leads to misinterpretation of the differences between strains. The lines in **Fig 1D** represent read coverage, they represent data better when many small RNAs map at the same locus. On the other hand, in **Fig 3A** there are too many samples to show as read coverage tracks and for this reason we use the most compact representation of the data per sample.

5) Figure 3B: The authors found a good correlation between "black" and "white" genotypes (according to the legend in Fig 3B) and the number of 19-36 nt small RNAs mapped to the Noct locus. However, it remains to be examined whether such a correlation can be applied to "gray"

genotype. This needs to be done.

We have now genotyped all ICR mouse samples (**Supplementary Fig 7**), except one mouse (sample 12) for which we had no remaining tissue in the freezer. The revised version of **Fig 3B** shows all the genotyped samples. The conclusion remains the same.

6) Figure 4A: The authors examined "clusters with indels" and "clusters without indels". However, the data should be heavily influenced by which strain of mice the author used as the reference for their analysis. For example, the Noct locus had an IAP insertion in BL6 but not in C3H. If the authors consider BL6 as a reference, C3H has an IAP insertion. If the authors consider C3H as a reference, BL6 has an IAP deletion. How are these handled?

We do not distinguish between deletions versus insertions. All analyses are done for strain pairs. The set of "clusters with indels" is specific to the pair of strains analysed. A cluster can be with indels in one strain pair comparison and without indels in a different strain pair comparison. In **Fig 4**, for each strain pair we test the association between the presence of indels and differential piRNA cluster expression. Our analysis is agnostic to whether an IAP is inserted in one strain or deleted in the other, as long as the IAP is present in one and not in the other strain at this position. A strain pair does not have an IAP indel when two strains either both have the IAP or both do not have it.

7) Figure 5A: The authors state that Noct mRNA expression is nearly identical among mouse strains, regardless of the insertion of IAP into the gene. However, Noct mRNA with IAP sequence is constantly consumed as "piRNA" precursors upon transcription. Therefore, it is necessary to examine the mRNA levels after piRNA biogenesis in the testes is completely inhibited from the first step (right after transcription).

The data shown in **Fig 5** show that Noct is expressed in strains with and strains without the IAP insertion. Our conclusion is limited to the fact that both Noct alleles produce similar steady state levels of precursor transcripts although the levels of piRNAs produced are significantly different. We believe that the suggested experiment inhibiting piRNA biogenesis is beyond the scope of this manuscript.

8) Page 5: "it looks like the IAP is causing the unspliced transcript to be exported from the nucleus and to be recognised as a piRNA precursor." I have heard of the phenomenon that inserting a TE sequence into a protein-coding mRNA can affect the splicing pattern of the host mRNA. However, the aberrant, unspliced mRNAs should be retained in the nucleus. It seems very unlikely to me that the insertion of IAP would result in "active" export of unspliced mRNA into the cytoplasm. This makes me doubt whether the authors' interpretation is correct. Also, does such splicing inhibition and promotion of pre-mRNA export by IAP insertion affect the expression level of NOCT protein? This needs to be examined.

In general, unspliced mRNAs are retained in the nucleus but there are exceptions. As cross-commented by reviewer #1, one such exception is the case of infectious retroviruses that require the active export of their unspliced transcripts to the cytoplasm for their survival (e.g. see (Ernst et al, 1997). Mouse IAPs are endogenous retroviruses, remnants of previously infectious retroviruses, and known for their ability to exit the nucleus as unspliced transcripts (see (Zolotukhin et al, 2008)). Our hypothesis is that the IAP insertion inside the intron of Noct leads to the export of unspliced, or aberrantly spliced transcripts to the cytoplasm. Furthermore, as commented by referee #1, there is already evidence of an association between piRNA production and lack of splicing.

Since in this manuscript we do not make any claims regarding protein levels or the phenotypic consequences of the IAP insertion in Noct, we do not think that examination of NOCT protein levels would be relevant. If the point for doing this additional analysis is to use it as evidence of whether there is a change in the amount of RNA exported from the nucleus, we still think that providing NOCT protein levels will not be helpful because, although genomewide protein and transcript levels correlate, their correlation is not high enough to be of predictive value (reviewed in (Liu et al, 2016)).

We have added an extra sentence in the discussion on the evidence from other papers on the link between intron retention and piRNA production:

“There is association between intron retention (and unusually long first exons) and piRNA production in both mammals and flies (Yu et al, 2021, 2019; Zhang et al, 2014).”

I suggest that the authors follow additional suggestions when revising the manuscript. Some of these may overlap with the suggestions made above.

- Exclude data from "predicted piRNA clusters"*

Since we are comparing piRNAs in different strains and the known piRNAs are from only one strain (C57BL6), we believe that the predicted clusters add value to the manuscript.

- Concentrate on four mice strains, BL6, NOD, C3H, and 129*

CAST is derived from a different subspecies and the strain most genetically different from the reference mouse strain. Therefore, we think that it is useful to include it.

- Show detailed mapping data for the five piRNA-producing loci in Figure 1C*

*In **Supplementary Fig 4** we show mapping data for all samples for the piRNA-producing loci shown in Figure 1C.*

- Show TE insertions in the five loci*

*TE insertions/deletions from Nellaker et al are included in **Supplementary Fig 4**.*

- *ChIP data is not required*

We have moved the original **Fig 5B**, which contains the ChIP-seq data, to **Supplementary Fig 8**.

- Do you have suggestions that would help the authors improve the presentation of their data and conclusions?

1) In Figure 1, known piRNA clusters are analyzed, and in Figure 2, new piRNA clusters are analyzed. I would like to see more information on the relationship between them. For example, how many of the de novo piRNA clusters identified in Figure 2 were known in Figure 1?

We have added a new panel (panel B) in **Fig 2**, in which we show the proportion of the predicted piRNA clusters that overlap previously known clusters.

2) Figure 2B: Please state that this is about a total of six pairwise comparisons made between four different testis strains.

Done.

3) Figures 2D and 2E: Please indicate what percentage of each type of piRNA cluster is present.

Using the automatic annotation of the predicted clusters by proTRAC (**Supplementary Table 5 and 6**), of the 900 predicted loci, 737 (82%) produce piRNAs from only one strand, like the one shown in **Fig 2E** (corresponding to panel D of the first version of the manuscript). The remaining 163 loci (18%) produce piRNAs from both strands, like the one shown in **Fig 2F**.

We have added these numbers to the 4th paragraph of the results:

“There are 900 predicted piRNA-producing loci, 82% of which produce piRNAs from only one strand and 18% from both strands.”

4) Figure 3A: Please describe exon, intron, UTR, splicing variant, and repeat (Figure 3A) in the text. What do brown piRNAs out of piRNA mapping indicate?

We have added the requested information to the legend of **Fig 3**:

“In the schematic at the top, coding exons are indicated as wide blue rectangles, UTRs are thinner blue rectangles, and introns as the lines connecting them. piRNA clusters are indicated as grey rectangles and polymorphic repeats as red rectangles. Small RNAs mapping to the same strand as the gene are shown as blue lines and those mapping to the antisense strand as red lines.”

Reviewer #2 (Significance (Required)):

Significance (Required)

- Describe the nature and significance of the advance (e.g. conceptual, technical, clinical) for the field.

The problems they tried to tackle in this study are important in the field of piRNA.

- Place the work in the context of the existing literature (provide references, where appropriate). It is well known in the field that the identity of piRNA clusters is acquired by Rhino, H3K9me3, etc. from fly analysis, and furthermore, that new insertion of piRNA is related to piRNA production. On the other hand, it is a unique approach and very interesting to compare this issue in mammal and by mouse subspecies.

- State what audience might be interested in and influenced by the reported findings. piRNA researchers. Embryologists with a focus on the testis.

- Define your field of expertise with a few keywords to help the authors contextualize your point of view. Indicate if there are any parts of the paper that you do not have sufficient expertise to evaluate.

Reproductive epigenome

- EIMaghraby MF, Andersen PR, Pühlinger F, Hohmann U, Meixner K, Lendl T, Tirian L & Brennecke J (2019) A Heterochromatin-Specific RNA Export Pathway Facilitates piRNA Production. *Cell* 178: 964-979.e20
- Ernst RK, Bray M, Rekosh D & Hammarskjöld M-L (1997) A Structured Retroviral RNA Element That Mediates Nucleocytoplasmic Export of Intron-Containing RNA. *Mol Cell Biol* 17: 135–144
- Kaaij LJT, Hoogstrate SW, Berezikov E & Ketting RF (2013) piRNA dynamics in divergent zebrafish strains reveal long-lasting maternal influence on zygotic piRNA profiles. *RNA* 19: 345–356
- Kelleher ES & Barbash DA (2013) Analysis of piRNA-Mediated Silencing of Active TEs in *Drosophila melanogaster* Suggests Limits on the Evolution of Host Genome Defense. *Mol Biol Evol* 30: 1816–1829
- Kneuss E, Munafò M, Eastwood EL, Deumer U-S, Preall JB, Hannon GJ & Czech B (2019) Specialization of the *Drosophila* nuclear export family protein Nxf3 for piRNA precursor export. *Genes Dev* 33: 1208–1220
- Li XZ, Roy CK, Dong X, Bolcun-Filas E, Wang J, Han BW, Xu J, Moore MJ, Schimenti JC, Weng Z, *et al* (2013) An Ancient Transcription Factor Initiates the Burst of piRNA Production during Early Meiosis in Mouse Testes. *Mol Cell* 50
- Liu Y, Beyer A & Aebersold R (2016) On the Dependency of Cellular Protein Levels on mRNA Abundance. *Cell* 165: 535–550

- Özata DM, Yu T, Mou H, Gainetdinov I, Colpan C, Cecchini K, Kaymaz Y, Wu P-H, Fan K, Kucukural A, *et al* (2020) Evolutionarily conserved pachytene piRNA loci are highly divergent among modern humans. *Nat Ecol Evol* 4
- Shpiz S, Ryazansky S, Olovnikov I, Abramov Y & Kalmykova A (2014) Euchromatic Transposon Insertions Trigger Production of Novel Pi- and Endo-siRNAs at the Target Sites in the *Drosophila* Germline. *PLoS Genet* 10: e1004138
- Yu T, Biasini A, Cecchini K, Säflund M, Mou H, Arif A, Eghbali A, de Rooij DG, Weng Z, Zamore PD, *et al* (2023) A-MYB/TCFL5 regulatory architecture ensures the production of pachytene piRNAs in placental mammals. *RNA* 29: 30–43
- Yu T, Fan K, Özata DM, Zhang G, Fu Y, Theurkauf WE, Zamore PD & Weng Z (2021) Long first exons and epigenetic marks distinguish conserved pachytene piRNA clusters from other mammalian genes. *Nat Commun* 12: 73
- Yu T, Koppetsch BS, Pagliarani S, Johnston S, Silverstein NJ, Luban J, Chappell K, Weng Z & Theurkauf WE (2019) The piRNA Response to Retroviral Invasion of the Koala Genome. *Cell* 179: 632-643.e12
- Zhang Z, Wang J, Schultz N, Zhang F, Parhad SS, Tu S, Vreven T, Zamore PD, Weng Z & Theurkauf WE (2014) The HP1 Homolog Rhino Anchors a Nuclear Complex that Suppresses piRNA Precursor Splicing. *Cell* 157: 1353–1363
- Zolotukhin AS, Schneider R, Uranishi H, Bear J, Tretyakova I, Michalowski D, Smulevitch S, O’Keeffe S, Pavlakis GN & Felber BK (2008) The RNA transport element RTE is essential for IAP LTR-retrotransposon mobility. *Virology* 377: 88–99

Dear Dr. Vavouri,

Thank you for submitting your manuscript for consideration by the EMBO Journal. It has now been seen by two referees whose comments are shown below. Referee #1 corresponds to referee #1 from the first round of review at Review Commons. Unfortunately referee #2 was not available and we have contacted an additional referee who is an expert in the field and familiar with The EMBO Journal. As you will see, this referee thinks that the topic as well as the analysis performed in the manuscript are interesting but also remarks that the manuscript would benefit from additional analysis.

We think that these requests are reasonable and we would therefore like to invite you to submit a revised version of the manuscript, addressing these comments. I should add that it is EMBO Journal policy to allow only a single round of revision, and acceptance of your manuscript will therefore depend on the completeness of your responses in this revised version.

Thank you for the opportunity to consider your work for publication. I look forward to your revision.

Yours sincerely,

Cornelius Schneider

Cornelius Schneider, PhD
Editor
The EMBO Journal
c.schneider@embojournal.org

We realize that it is difficult to revise to a specific deadline. In the interest of protecting the conceptual advance provided by the work, we recommend a revision within 3 months (24th Dec 2024). Please discuss the revision progress ahead of this time with the editor if you require more time to complete the revisions. Use the link below to submit your revision:

Referee #2:

I reviewed this manuscript on the basis of reviewer comments on Review Commons and a point-by-point response, but, as instructed, also carried out my own review.

Summary:

Here the authors investigated piRNA variability across mouse strains using small RNA sequencing from 4 mouse lines and an outbred line. They find differences in expression between the line and link some of these to genetic differences. Some of these genetic differences match up with transposable element insertions that are different between the strains, which supports the existing model for piRNA cluster evolution whereby transposable element insertions have a large role in driving rapid diversification.

Strengths:

The manuscript reports a very useful dataset for the community. The analyses are very well described in the excellent materials and methods section, and the choices that the authors made for processing and aligning the data etc set a standard for best practice in the field, particularly as the requirement to align to several different strain-specific genomes made this a non-trivial task. The authors have also prepared a solid response to the Review Commons reviews, and I was satisfied that they had dealt with the queries by Reviewer 2 in particular in a thorough way improving the manuscript considerably.

Weaknesses

The conclusion of the paper at present is somewhat weak and does not provide as much of an advance to the field as it could, given the dataset. Fundamentally, it is not at all surprising that genetic differences result in differences in piRNAs and this conclusion on its own is not really of wide interest. The key discovery that this paper could make is to demonstrate that the genetic variation between strains is THE MOST IMPORTANT aspect leading to diversity in piRNAs- in particular whether quantitatively there is more variation between strains than would be expected given individual variability. The authors make a brief claim to this extent but do not emphasize it or assess it quantitatively (it is only a qualitative evaluation in Fig 1B and Sup fig 3). I would really think that the authors ought to better substantiate this claim to make their manuscript enough of an advance for it to be suitable for the EMBOJ. I have three analyses that I would suggest that the authors could include that would get at this (see specific suggestion section).

Another weakness that ought to be acknowledged is that the authors only look at the 4 strains. I don't suggest that they need to look at more at this stage but I would like to see more reference to the fact that this is actually quite a small number and that there are more inbred strains available and so eventually one would expect further analyses to supersede the current manuscript. Perhaps more acknowledgement of this in the discussion would be helpful.

Specific suggestions

1. Key is to show statistically that the strain is a significant cause of variability in piClust expression above the individual differences. I suggest that a linear model would be a good way to approach this. Two options could be tried. One would be to consider all clusters together with a model of the form $\text{expression} \sim (\text{strain} + \text{individual}) | \text{cluster}$ where cluster is a random effect and strain and individual are fixed effects. This would provide a global assessment. Is strain a significant effector? A second, more interesting approach would be to do a model for each cluster, potentially an anova of the form $\text{expression} \sim \text{strain} + \text{individual}$ (two way anova). Then you could identify, after multiple test correction, which clusters have significant variation linked to strain. Which fraction show this? Are these the ones that in general tend to be most variable in terms of the number of snps/kb? It would also allow the clusters where there is considerable inter-individual variation to be

identified and these compared to those where there is strain variability. Are these the same or are some highly similar clusters showing large inter-individual variability, which would be very interesting (also see below).

2) The authors can ask whether the overall difference between strains in piRNA expression correlates to the genetic distance or not. With only 4 strains this is not going to be very rigorous but at least the authors could say that the genetic tree matches with the expression tree.

Together these two analyses would substantiate the main point of the manuscript and would be very interesting to see.

3) A separate but linked analysis would be to take advantage of the outbred strains to identify the most variable clusters. Here, with 39 individuals, the authors could plot a classic 'noise' plot ie $(CV)^2$ vs mean and fit a curve, thus identifying through residuals which clusters show more inter-individual noise and which show less inter-individual noise. Using data on snp density in ICR mice (ie polymorphic sites) they could then correlate genetic distance with noise in genetically non-distinct individuals. This would be really nice and take advantage of the rich dataset provided by the work.

Referee #3:

The authors have fully satisfied my concerns, and I look forward to seeing the manuscript published as soon as possible.

Manuscript ref: EMBOJ-2024-118473R

Corresponding author(s): Tanya, Vavouri; Sonia, Forcales; Josep C., Jimenez-Chillaron

General Statement

We appreciate the feedback provided by the reviewer. Their insightful comments have helped us improve the manuscript. Here, we address each comment and detail the corresponding modifications made in the revised manuscript.

Referee #2:

I reviewed this manuscript on the basis of reviewer comments on Review Commons and a point-by-point response, but, as instructed, also carried out my own review.

Summary:

Here the authors investigated piRNA variability across mouse strains using small RNA sequencing from 4 mouse lines and an outbred line. They find differences in expression between the line and link some of these to genetic differences. Some of these genetic differences match up with transposable element insertions that are different between the strains, which supports the existing model for piRNA cluster evolution whereby transposable element insertions have a large role in driving rapid diversification.

Strengths:

The manuscript reports a very useful dataset for the community. The analyses are very well described in the excellent materials and methods section, and the choices that the authors made for processing and aligning the data etc set a standard for best practice in the field, particularly as the requirement to align to several different strain-specific genomes made this a non-trivial task. The authors have also prepared a solid response to the Review Commons reviews, and I was satisfied that they had dealt with the queries by Reviewer 2 in particular in a thorough way improving the manuscript considerably.

We are glad to read that the reviewer was satisfied with our response to the comments of the two reviewers from Review Commons.

Weaknesses

The conclusion of the paper at present is somewhat weak and does not provide as much of an advance to the field as it could, given the dataset. Fundamentally, it is not at all surprising that genetic differences result in differences in piRNAs and this conclusion on its own is not really of wide interest. The key discovery that this paper could make is to demonstrate that the genetic variation between strains is THE MOST IMPORTANT aspect leading to diversity in piRNAs- in particular whether quantitatively there is more variation between strains than would be expected given individual variability. The authors make a brief claim to this extent but do not emphasize it or assess it

quantitatively (it is only a qualitative evaluation in Fig 1B and Sup fig 3). I would really think that the authors ought to better substantiate this claim to make their manuscript enough of an advance for it to be suitable for the EMBOJ. I have three analyses that I would suggest that the authors could include that would get at this (see specific suggestion section).

Another weakness that ought to be acknowledged is that the authors only look at the 4 strains. I don't suggest that they need to look at more at this stage but I would like to see more reference to the fact that this is actually quite a small number and that there are more inbred strains available and so eventually one would expect further analyses to supersede the current manuscript. Perhaps more acknowledgement of this in the discussion would be helpful.

We thank the reviewer for identifying these weaknesses. We have now addressed these two points in the discussion.

Specifically, having carried out several new analyses to address the reviewer's comments and having now quantitatively shown that, for some piRNA clusters, genetics is indeed the most important aspect leading to piRNA cluster variation in genetically different mice, we have added the following sentence to the first paragraph of the discussion:

“In particular, for some piRNA-producing loci we found that genetic differences explain most of the variation in expression between different mice.”

Regarding the small number of inbred strains used in our paper, we have commented on this limitation in the last paragraph of the discussion:

“Here, we analysed piRNAs in a handful of mouse strains but there are many more mouse strains with sequenced genomes that could be used to study piRNA expression variation in this mammal.”

Specific suggestions

1. Key is to show statistically that the strain is a significant cause of variability in piClust expression above the individual differences. I suggest that a linear model would be a good way to approach this. Two options could be tried.

One would be to consider all clusters together with a model of the form $\text{expression} \sim (\text{strain} + \text{individual}) / \text{cluster}$ where cluster is a random effect and strain and individual are fixed effects. This would provide a global assessment. Is strain a significant effector?

In our dataset we have several strains and several individuals per strain, but only one sample per individual, so we can estimate the strain as a fixed effect but not the effect of the individual which is therefore included in the residual error. To assess the statistical significance of strain on piRNA cluster expression we compared the linear model with strain as a fixed effect and the cluster as a random effect, against the simpler model without the strain effect. The expression values were scaled by DESeq2 and transformed using the Variance Stabilizing Transformation. We then compared the two models with ANOVA to test whether the model with strain fits the data significantly better. We found that the model that includes strain fits the data significantly better with a p-value of 0.04. Therefore, globally, strain is a significant cause of variability in piRNA cluster expression.

We have added this result in the second paragraph of the Results section:

“To assess statistically the effect of the strain on global piRNA cluster expression, we used a linear model with strain as a fixed effect and piRNA cluster as a random effect and found that strain significantly contributes to piRNA cluster expression level (p-value = 0.04)”

A second, more interesting approach would be to do a model for each cluster, potentially an anova of the form $expression \sim strain + individual$ (two way anova). Then you could identify, after multiple test correction, which clusters have significant variation linked to strain. Which fraction show this? Are these the ones that in general tend to be most variable in terms of the number of snps/kb? It would also allow the clusters where there is considerable inter-individual variation to be identified and these compared to those where there is strain variability. Are these the same or are some highly similar clusters showing large inter-individual variability, which would be very interesting (also see below).

For each cluster, we fit a model with strain as a fixed effect and found that for 83% (159 out of 192) of the clusters, strain significantly explains variation between samples (Benjamini-Hochberg adjusted p-value < 0.05) and that for 66% (127/192) of the clusters, strain explains 80% of the variation. The adjusted R-squared and adjusted p-values can be found in **Expanded View Table EV4**.

We have edited the second paragraph of the Results section to include this analysis:

“Considering each cluster independently, strain significantly contributes to inter-individual variation in piRNA cluster expression in 159 out of 192 (83%) clusters (Table EV4). Notably, strain explains at least 80% of inter-individual variation in expression in 127 out of 192 (66%) the clusters (Fig 1C). We conclude that in classical inbred strains, for a subset of piRNA clusters, genetics is the most important factor affecting inter-individual variation of piRNA expression.”

We added the methodology related to the linear models in the Methods section under the heading “Differential expression analysis of piRNA producing loci and test of association with variable transposable elements”:

“We used DESeq2 v1.34.0 (Love et al, 2014) to normalize read counts with respect to library size. To study quantitatively the effect of genetics on piRNA cluster expression using linear models, we used the Variance Stabilising Transformation (VST) (Anders & Huber, 2010) to transform the piRNA cluster expression values. To assess the effect of strain on piRNA cluster expression globally, we used the lme4 package (Bates et al, 2015) and modelled strain as a fixed effect and piRNA cluster as a random effect ($expression \sim strain + (1|cluster)$). To assess the effect of strain on piRNA cluster expression for each cluster separately, we tested whether strain as a fixed effect explains a significant fraction of inter-individual piRNA cluster expression variation ($expression \sim strain$). The adjusted R-squared, the nominal p-value and the p-value adjusted with the Benjamini Hochberg correction (Benjamini & Hochberg, 1995) are shown in Table EV4.”

Regarding the question on whether piRNA clusters with high variation in expression tend to be the most variable, in our manuscript we propose that endogenous retrovirus insertions have a big impact on piRNA production. Although we expect that other types of polymorphisms also affect piRNA cluster expression, we have no reason to believe the number of SNPs or SNP density should be proportional to piRNA expression variation, especially when analysing only four classical inbred strains generated through artificial breeding.

2) The authors can ask whether the overall difference between strains in piRNA expression correlates to the genetic distance or not. With only 4 strains this is not going to be very rigorous but at least the authors could say that the genetic tree matches with the expression tree. Together these two analyses would substantiate the main point of the manuscript and would be very interesting to see.

For each piRNA cluster, we used known polymorphisms (dbSNP142) between the 4 strains to calculate the genetic distance tree and compared it to the expression distance tree. Of the 192 piRNA clusters, 128 have at least 10 informative SNPs that we used to calculate a gene tree. We built a tree based on the euclidian distance and compared

the genetic distance tree with the expression distance tree using the Robinson-Foulds distance and the Adjusted Rand Index (**Expanded View Table EV6**). In **Appendix Figure 5**, we show the genetic and the expression tree for the piRNA clusters shown in **Figure 1E**. For cluster 10-qC1-2617 there is only 1 SNP overlapping it, so no genetic tree was calculated. For pi-Noct and pi-Mrs2, the genetic and expression trees match. For pi-Zbtb37 two strains cluster together in both trees while the other strains do not. Considering all piRNA clusters together, we did not find that the expression and the genetic trees matched significantly more often than expected by chance. As mentioned by the referee and in our previous response, this analysis is limited by the low number of strains, the artificial breeding of the strains and the fact that we do not expect that the genetic distance based on the number of SNPs along the gene is proportional to the expression distance of the cluster. We address the link between genetics and gene expression variation using the outbred strain.

We have edited the text to include this result:

“Using known SNPs between the four inbred mouse strains, for each cluster, we compared the genetic distance tree to the expression distance tree. The genetic and expression trees match for pi-Noct and pi-Mrs2, they match partially for pi-Zbtb37 while the non-coding piRNA cluster 10-qC1-2617 does not contain enough SNPs to calculate a genetic distance tree (see Appendix Fig 5 and Table EV6).”

We added the methodology of the generation of the genetic and expression distance trees in the Methods section under the heading “Differential expression analysis of piRNA producing loci and test of association with variable transposable elements”:

“To compare the genetic distance to the expression distance between the four classical inbred strains, for each piRNA cluster separately, we calculated the genetic (euclidean) distance between the four strains using known SNPs (dbSNP142) overlapping the cluster, also calculated the expression (euclidean) distance using the VST transformed piRNA cluster expression values and compared the tree of the genetic and expression distances using the Robinson-Foulds metric and the Adjusted Rand Index matrix (Table EV6).”

3) A separate but linked analysis would be to take advantage of the outbred strains to identify the most variable clusters. Here, with 39 individuals, the authors could plot a classic 'noise' plot ie $(CV)^2$ vs mean and fit a curve, thus identifying through residuals which clusters show more inter-individual noise and which show less inter-individual noise. Using data on snp density in ICR mice (ie polymorphic sites) they could then correlate genetic distance with noise in genetically non-distinct individuals. This would be really nice and take advantage of the rich dataset provided by the work.

The reviewer suggests a correlation test between expression distance and genetic distance for the outbred animals in our study. As we do not have the genomes or polymorphisms of the 39 animals, we cannot use SNP density. However, we have information about their relatedness, which is also a measure of genetic distance between individuals in a population. We followed the method that the reviewer suggested to identify piRNA clusters with high expression “noise” in the 39 ICR mice. The plot of the coefficient of variation squared versus the mean is shown in **Appendix Figure 9**. The five piRNA clusters ranked with the highest noise in expression are highlighted in red in **Appendix Figure 9A** and their expression values (adjusted for library size, transformed with VST and batch-corrected) are shown in **Appendix Figure 9B**. We then used the relatedness values from the pedigrees to estimate the percentage of variation explained by genetics (i.e. the heritability of piRNA expression level). Overall, the average heritability estimate of piRNA cluster expression was 48%. Because the pedigrees are incomplete (we only have measurements for males) and because the dataset is relatively small, to estimate the significance of the contribution of genetics on piRNA cluster expression level, we used 100 permutations of the individuals in the pedigrees and for each permutation we repeated the estimation of heritability. From the permutations we calculated the probability of getting the same or higher heritability estimate by chance and adjusted these p-values to account for multiple testing using the Benjamini-Hochberg method. We found that genetics significantly contributes to at

least 50% of variation in piRNA cluster expression for 87 out of the 214 clusters (40.7%). For 3 of the 5 most “noisy” piRNA clusters, the estimated heritability of expression level was higher than in 95% of the permuted pedigrees (heritability estimates and adjusted p-values are shown in **Expanded View Table 8**). This result provides independent evidence that for a subset of piRNA producing loci genetics explains the majority of inter-individual expression.

We have included this analysis in the first section of the results:

“To gain independent evidence for the link between genetic variation and gene expression variation we sequenced and analysed small RNAs from testes of 39 young adult mice of the genetically outbred strain ICR (for further details on this dataset see Methods and Table EV1, Table EV7). These mice were from seven pedigrees (Appendix Fig 8). If genetics has a major effect on inter-individual piRNA cluster expression, then, at least for some piRNA clusters, closely related individuals should have more similar piRNA cluster expression than less related individuals. For each piRNA cluster, we tested whether relatedness is a factor that significantly explains piRNA expression variation. We found that on average 48% of variation of piRNA cluster expression is explained by relatedness with a distribution skewed towards high values (Fig 1D, Table EV8). Because the dataset is relatively small and the information incomplete (we only have data for males), we used permutations of individuals in pedigrees to assess the significance of the estimates of heritability of piRNA cluster expression. For each piRNA cluster, we compared the variance explained by genetics calculated from individuals in their true pedigrees against that from 100 permutations of the pedigrees. We found that genetics significantly contributes to at least 50% of variation in piRNA cluster expression for 87 out of the 214 clusters (40.7%). We wondered whether genetic differences and different degrees of relatedness also explain variation in piRNA cluster expression for the top five clusters that vary the most in this set. For three out of five of these piRNA clusters, genetics significantly explains most of the inter-individual piRNA cluster expression (Appendix Fig 9). The results from both inbred and outbred mice, collectively demonstrate that for some piRNA clusters, genetics explains most of the inter-individual expression variation. ”

Also we added the following sentence in the 2nd paragraph of the section entitled “Association between an intronic IAP insertion and piRNA production from the mouse protein-coding gene Nocturnin”:

“Furthermore, 81.4% of pi-Noct expression variation is due to genetics, based on the analysis of the effect of genetic relatedness on piRNA cluster expression in mouse pedigrees of the outbred strain ICR (Fig 3C, Table EV8).”

We added the methodology of the estimation of the percentage of expression variance due to genetics in ICR mice in the Methods section under the heading “Differential expression analysis of piRNA producing loci and test of association with variable transposable elements”:

“For the estimation of heritability of piRNA cluster expression in the samples of the ICR strain, because the samples were sequenced in two batches we batch-corrected the library size adjusted and VST transformed piRNA cluster expression values using the sva R package (Leek et al, 2012). For the estimation of piRNA cluster expression heritability, we first calculated the Additive Relationship Matrix using pedigreemm package (Vazquez et al, 2010) from the mouse pedigrees (Appendix Fig 8). To estimate heritability we used the lme4breeding R package (Covarrubias-Pazarán, 2024), where expression was modelled with the individual as a random effect linked to the Additive Relationship Matrix and the pedigree set as a fixed effect (three pedigrees belong to one set and the other four in a different set, where the male ancestors of the second set were overnursed). The variance of piRNA cluster expression due to genetics was also calculated in 100 permuted pedigrees, where animals of the same generation and the same pedigree set were shuffled, maintaining animals in the same generation and the same pedigree set.”

Referee #3:

The authors have fully satisfied my concerns, and I look forward to seeing the manuscript published as soon as possible.

Dear Dr Vavouri,

Thank you for submitting a revised version of your manuscript. Your study has now been seen by all original referees, who find that their previous concerns have been addressed and now recommend publication of the manuscript. There remain only a few mainly editorial points that have to be addressed before I can extend formal acceptance of the manuscript:

- On the abstract page of the manuscript, please include 4-5 general keyword terms to enhance searchability.
 - Please rename DAS to Data Availability and placed before Acknowledgments
 - Please rename the Conflict-of-Interest section into "Disclosure and Competing Interests Statement", in accordance with our updated Guide to Authors (<https://www.embopress.org/competing-interests>)
 - Please place the authors' emails on the title page of the ms. There is also a name discrepancy - J. Andrew Pospisilik in the ms vs. Andrew J. Pospisilik in eJP
 - As we are switching from a free-text author contribution statement towards a more formal statement based on Contributor Role Taxonomy (CRediT) terms, please remove the present Author Contribution section and instead specify each author's contribution(s) directly in the Author Information page of our submission system during upload of the final manuscript. See <https://casrai.org/credit/> for more information.
 - There is a reference to "data not shown" on page 9, "Differential expression analysis of piRNA producing loci and test of association with variable transposable elements" section. According to our policy, which does not permit references to "data not shown", please include this information in the Appendix. Please see also <https://www.embopress.org/page/journal/14602075/authorguide#unpublisheddata>.
 - Please double-check to make sure to all relevant funding information in the manuscript is also entered into our submission system. (Missing in the system currently: CERCA Programme/Generalitat de Catalunya; ID 100010434)
 - Please adjust the in-text callouts for individual figures and figure panels: e.g. Fig 6A appears to be missing and callout "EV Table 1" is not a correct format (see below)
 - DATASET EV LEGENDS: on Excel file with 20 EV tables uplidd; all these tables except for Table EV12 are datasets as they have multiple rows and columns; each table needs to be uplidd as a separate Excel file and the correct nomenclature (source file names, titles in eJP, legends, ms callouts) should be Dataset EV1- Dataset EV20 (or Dataset EV19 if Table EV12 is going to remain as an EV table in which case it should be Table EV1); the legends need to be removed from the ms file and need to be provided in each Excel file as a separate sheet/tab
 - APPENDIX FILE WITH ToC: in, but the nomenclature is not consistent and correct in all places; it should be Appendix Figure S# everywhere (in the Appendix file, in the figure titles throughout the file in the ms callouts); also, there shouldn't be Appendix Figure S3A, Appendix Figure S3B, etc. - all panels of one figure should be combined into one figure - Appendix Figure S3; the legends need to be removed from the ms file and each needs to be located right after its corresponding figure in the Appendix file
 - Please provide suggestions for a short 'blurb' text prefacing and summing up the conceptual aspect of the study in two sentences (max. 250 characters), followed by 3-5 one-sentence 'bullet points' with brief factual statements of key results of the paper; they will form the basis of an editor-written 'Synopsis' accompanying the online version of the article. Please also provide an altered synopsis image, making sure that the aspect ratio conforms to our website's format - it should be exactly 550 pixels wide and between 300-600 pixels high.
 - Please place the Main Figure legends at the end of the ms, after the References
 - Please rename the Materials and Methods to Methods
 - Please provide the specific URL for GSE215030 dataset in the data availability statement.
 - Figure Legends - Comments
1. Please note that the exact p values are not provided in the legends of figures 4A, 6A.
 2. Please indicate the statistical test used for data analysis in the legends of figures 1D, 6B.

3. Please note that the box plots need to be defined in terms of minima, maxima, centre, bounds of box and whiskers, and percentile in the legends of figures 1C, 5A.

4. Please note that information related to n is missing in the legends of figures 1C, 4A, B; 5A, 6A, B.

5. Please note that the error bars are not defined in the legends of figures 4A, 6A, B.

With best regards,

Cornelius Schneider

Cornelius Schneider, PhD
Editor
The EMBO Journal
c.schneider@embojournal.org

We realize that it is difficult to revise to a specific deadline. In the interest of protecting the conceptual advance provided by the work, we recommend a revision within 3 months (20th May 2025). Please discuss the revision progress ahead of this time with the editor if you require more time to complete the revisions. Use the link below to submit your revision:

Referee #2:

I thank the authors for their thorough response to my comments. I'd be delighted to see this study published.

Manuscript ref: EMBOJ-2024-118473R

Corresponding author(s): Tanya, Vavouri; Sonia, Forcales; Josep C., Jimenez-Chillaron

General Statement

We thank the reviewer and the editor for recommending our manuscript for publication. Here, we list the changes made in response to the editorial comments:

On the abstract page of the manuscript, please include 4-5 general keyword terms to enhance searchability.

We have added the keywords: ERV, diversity, mouse, piRNA, PIWI

Please rename DAS to Data Availability and placed before Acknowledgments

Done

Please rename the Conflict-of-Interest section into "Disclosure and Competing Interests Statement", in accordance with our updated Guide to Authors (<https://www.embopress.org/competing-interests>)

Done

Please place the authors' emails on the title page of the ms. There is also a name discrepancy - J. Andrew Pospisilik in the ms vs. Andrew J. Pospisilik in eJP

Done

As we are switching from a free-text author contribution statement towards a more formal statement based on Contributor Role Taxonomy (CRediT) terms, please remove the present Author Contribution section and instead specify each author's contribution(s) directly in the Author Information page of our submission system during upload of the final manuscript. See <https://casrai.org/credit/> for more information.

Done

There is a reference to "data not shown" on page 9, "Differential expression analysis of piRNA producing loci and test of association with variable transposable elements" section. According to our policy, which does not permit references to "data not shown", please include this information in the Appendix. Please see also <https://www.embopress.org/page/journal/14602075/authorguide#unpublisheddata>.

To show that the conclusions are not affected by the inclusion or exclusion of reads mapping to repeats, in the main text we show the results including reads mapping to annotated repeats (**Expanded View Tables EV5, EV9, EV10**) and we have now added the corresponding tables showing the results excluding reads mapping to repeats in **Expanded View Tables EV21, EV22 and EV23**. Note that we have corrected a

mistake in the description of the methodology on this point: in the previous version of the manuscript we had written that the results shown in the manuscript were those excluding reads mapping to repeats but they were the results including reads mapping to repeats. As already explained, the inclusion or exclusion of repeats has minimal effect on the results and now the reader can see the results of both methods.

Please double-check to make sure to all relevant funding information in the manuscript is also entered into our submission system. (Missing in the system currently: CERCA Programme/Generalitat de Catalunya; ID 100010434)

We have added two more sources of funding to the manuscript and to the online system. The “ID 100010434” is part of the funding acknowledgment to La Caixa Foundation.

Please adjust the in-text callouts for individual figures and figure panels: e.g. Fig 6A appears to be missing and callout "EV Table 1" is not a correct format (see below)

Done.

DATASET EV LEGENDS: on Excel file with 20 EV tables upldd; all these tables except for Table EV12 are datasets as they have multiple rows and columns; each table needs to be upldd as a separate Excel file and the correct nomenclature (source file names, titles in eJP, legends, ms callouts) should be Dataset EV1- Dataset EV20 (or Dataset EV19 if Table EV12 is going to remain as an EV table in which case it should be Table EV1); the legends need to be removed from the ms file and need to be provided in each Excel file as a separate sheet/tab

We have made a separate Excel file for each table and put the table title on the first tab of each file. We have maintained Table EV12 as **Expanded View Table EV12**. The legends have been removed from the manuscript file.

APPENDIX FILE WITH ToC: in, but the nomenclature is not consistent and correct in all places; it should be Appendix Figure S# everywhere (in the Appendix file, in the figure titles throughout the file in the ms callouts); also, there shouldn't be Appendix Figure S3A, Appendix Figure S3B, etc. - all panels of one figure should be combined into one figure - Appendix Figure S3; the legends need to be removed from the ms file and each needs to be located right after its corresponding figure in the Appendix file

Done.

Please provide suggestions for a short 'blurb' text prefacing and summing up the conceptual aspect of the study in two sentences (max. 250 characters), followed by 3-5 one-sentence 'bullet points' with brief factual statements of key results of the paper; they will form the basis of an editor-written 'Synopsis' accompanying the online version of the article. Please also provide an altered synopsis image, making sure that the aspect ratio conforms to our website's format - it should be exactly 550 pixels wide and between 300-600 pixels high.

Synopsis:

PIWI-interacting RNAs (piRNAs) are fast-evolving small RNAs expressed in the germline. This study reveals variation in locus-specific piRNA production between mouse strains, associated with endogenous retrovirus insertions.

- Different mouse strains express piRNAs at different levels and some piRNAs are strain-specific.
- Genetic variation is a major determinant of piRNA expression differences between individuals.
- Endogenous retrovirus insertions are associated with polymorphic expression of piRNAs.
- The insertion of an IAP endogenous retrovirus in the intron of protein-coding gene *Noct*, is associated with piRNA production from the locus in mouse strains that carry the insertion.

We have uploaded an image for the synopsis.

Please place the Main Figure legends at the end of the ms, after the References

Done.

Please rename the Materials and Methods to Methods

Done.

Please provide the specific URL for GSE215030 dataset in the data availability statement.

Done.

Figure Legends - Comments

Please note that the exact p values are not provided in the legends of figures 4A, 6A.

Because there are too many p-values to include in the legend, we have put them in three new Tables. The p-values of **Figure 4A** are in **Table EV1**, the p-values of **Figure 6A** upper panel are in **Table EV2** and the p-values of **Figure 6A** lower panel are in **Table EV3**. We refer to these tables in the legends of the figures.

Please indicate the statistical test used for data analysis in the legends of figures 1D, 6B.

Figure 1D: we have clarified in the legend that “Empirical p-values are based on 100 permutations and adjusted for multiple testing using the Benjamini-Hochberg method (see Methods)”.

Figure 6B: we have clarified in the legend that it is the Wilcoxon rank-sum test.

Please note that the box plots need to be defined in terms of minima, maxima, centre, bounds of box and whiskers, and percentile in the legends of figures 1C, 5A.

Figure 1C: we have clarified in the legend that “The boxplot inside the violin plot shows the interquartile range, the middle point corresponds to the median, and the whiskers extend to the extreme values provided they are no more than 1.5 times the interquartile range from the box”.

Figure 5A: we have clarified in the legend that “The boxplots show the interquartile range, the middle line corresponds to the median, and the whiskers extend to the extreme values provided they are no more than 1.5 times the interquartile range from the box”.

Please note that information related to n is missing in the legends of figures 1C, 4A, B; 5A, 6A, B.

Figure 1C, we have added “N = 191 piRNA-producing loci”.

Figure 4A, we have added “N = 749 predicted piRNA clusters.”

Figure 4B, we have added “N = 341 (no TEVs), 21 (LINEs in BL6), 7 (LINEs in CAST), 86 (SINEs in BL6), 35 (SINEs in CAST), 35 (ERVs in BL6), 39 (ERVs in CAST), 24 (IAPs in BL6). 17 (IAPs in CAST).”

Figure 5A, for the boxplots, we have added that “The number of genes in each boxplot is 19,796.”

Figure 6A, we have added “N = 749 predicted piRNA clusters.”

Figure 6B, we have added “N=29 (ERV on sense strand in BL6), 19 (IAP on sense strand in BL6), 10 (ERV on sense strand in CAST), 4 (IAP on sense strand in CAST), 21 (ERV on antisense strand in BL6), 14 (IAP on antisense strand on BL6), 12 (ERV on antisense strand in CAST), 6 (IAP on antisense strand in CAST).”

Please note that the error bars are not defined in the legends of figures 4A, 6A, B.

There are no error bars in any of these figures. In Figures Fig 4B and 6B, the lines indicate the group means, as already stated in the legend.

Other corrections:

- In the first sentence of the penultimate paragraph of page 4, in the previous version we had mistakenly referred to **Table EV4** instead of **Table EV5**. This has been corrected.
- In the previous version of **Table EV1**, some of the values of the column “AncestorOverfed” were mixed up. This has no impact on any of the results or figures or any part of the text. The values of the column have now been corrected.
- In **Appendix Figure S7C**, we had mistakenly labelled the top three plots as showing AMYB data and the bottom three plots as showing TCFL5 data, while it was the other way round. We have now corrected this.

Dear Dr. Vavouri,

I am pleased to inform you that your manuscript has been accepted for publication in the EMBO Journal.

Yours sincerely,

Cornelius Schneider, PhD
Editor
The EMBO Journal
c.schneider@embojournal.org
